# Glutamyl-prolyl-tRNA synthetase 1 coordinates early endosomal anti-inflammatory AKT signaling

Eun-Young Lee [1]✉, Su-Man Kim [1], Jung Hwan Hwang[2], Song Yee Jang[1,3], Shinhye Park[1], Sanghyeon Choi[1], Ga Seul Lee[3], Jungwon Hwang [1], Jeong Hee Moon[3], Paul L. Fox[4], Sunghoon Kim [5], Chul-Ho Lee [2] & Myung Hee Kim [1]✉

The AKT signaling pathway plays critical roles in the resolution of inflammation. However, the underlying mechanisms of anti-inflammatory regulation and signal coordination remain unclear. Here, we report that anti-inflammatory AKT signaling is coordinated by glutamyl-prolyl-tRNA synthetase 1 (EPRS1). Upon inflammatory activation, AKT specifically phosphorylates Ser999 of EPRS1 in the cytoplasmic multi-tRNA synthetase complex, inducing release of EPRS1. EPRS1 compartmentalizes AKT to early endosomes via selective binding to the endosomal membrane lipid phosphatidylinositol 3-phosphate and assembles an AKT signaling complex specific for anti-inflammatory activity. These events promote AKT activation-mediated GSK3β phosphorylation, which increase anti-inflammatory cytokine production. EPRS1-deficient macrophages do not assemble the early endosomal complex and consequently exacerbate inflammation, decreasing the survival of EPRS1-deficient mice undergoing septic shock and ulcerative colitis. Collectively, our findings show that the housekeeping protein EPRS1 acts as a mediator of inflammatory homeostasis by coordinating compartment-specific AKT signaling.

Inflammation is a dynamic response of the immune system that is indispensable for sustaining physiological homeostasis[1]. In response to inflammatory challenges, immune cells produce cytokines to trigger systemic responses that ultimately restore homeostasis[2]. Unresolved inflammation is a key driver of many disorders, including sepsis, cardiovascular diseases, type 2 diabetes, rheumatoid arthritis, and cancers[3–6].

Resolution of inflammation is a well-orchestrated process that involves the temporally and spatially controlled production of pro-inflammatory and anti-inflammatory cytokines by a variety of mediators. Macrophages are the major effector cells that facilitate innate immunity during acute inflammation and subsequent recovery[7]. During the inflammatory process, macrophages help to prevent prolonged inflammatory responses by producing anti-inflammatory cytokines such as IL-10[8]. Because the activity of macrophages can become detrimental under inflammatory conditions, tight regulation of sequential changes in macrophage functionality is essential for the recovery of homeostasis[9].

[1]Microbiome Convergence Research Center, Korea Research Institute of Bioscience and Biotechnology (KRIBB), Daejeon 34141, Korea. [2]Laboratory Animal Resource Center, KRIBB, Daejeon 34141, Korea. [3]Core Research Facility & Analysis Center, KRIBB, Daejeon 34141, Korea. [4]Department of Cardiovascular and Metabolic Sciences, Lerner Research Institute, Cleveland Clinic Foundation, Cleveland, OH 44195, USA. [5]Medicinal Bioconvergence Research Center, College of Pharmacy and College of Medicine, Gangnam Severance Hospital, Yonsei University, Incheon 21983, Korea. ✉e-mail: krupi00@kribb.re.kr; mhk8n@kribb.re.kr

Diverse molecular mechanisms underlie the resolution of inflammation. The phosphoinositide 3-kinase (PI3K)/AKT pathway mediates extracellular and intracellular signals that preserve the integrity of the immune system[10], particularly in macrophages[8]. As a negative feedback mechanism for Toll-like receptor (TLR) signaling, PI3K/AKT signaling restricts pro-inflammatory responses and promotes anti-inflammatory responses in LPS-stimulated macrophages by controlling the expression of miRNAs targeting TLR4 and SOCS1[11]. In addition, the AKT signaling pathway regulates TLR4 hypersensitivity in the myeloid lineage by influencing various cellular functions: it negatively regulates the MYD88-dependent NF-κB pathway in human monocytes to inhibit the expression of inflammatory mediators[12,13] and also modulates LPS-induced inflammatory responses by regulating the downstream target glycogen synthase kinase 3β (GSK3β). LPS-stimulated AKT phosphorylates GSK3β at Ser9, leading to its auto-inhibition, and subsequently decreases inflammatory cytokine production in monocytes and adipocytes[14,15].

Because the PI3K/AKT pathway regulates multiple downstream cellular processes, this axis must be elaborately regulated to target specific functions[16,17]. This sophisticated regulation is facilitated in part through compartment-specific activation of AKT[18,19]. Once activated (i.e., phosphorylated), AKT translocates to the plasma membrane as well as intracellular membrane compartments called organelles[20]. In particular, early endosomal membranes are ideally suited to act as specialized signaling platforms due to their distinct composition: they are enriched in mono-phosphoinositides, especially phosphatidylinositol 3-phosphate [PI(3)P], which promotes selective recruitment of scaffold proteins and signaling mediators[21]. For example, APPL1 recruits activated AKT to early endosomes to regulate receptor tyrosine kinase (RTK)-mediated MAPK/AKT signaling in macrophages and adipocytes[15,20,22]. Endosomal APPL1 signaling cannot control the lysosomal AKT–TSC2–mTORC1 pathway, highlighting the compartment-specific regulation of AKT signaling[20,22]. PTEN, which antagonizes PI3K/AKT signaling, also associates with PI(3)P-positive endosomes to terminate AKT activation[23]. However, these findings focused mainly on AKT activated in response to metabolic cues. Although the PI3K/AKT pathway plays a critical role in the resolution of inflammation via anti-inflammatory regulation[12], it remains unclear whether this form of AKT signaling is also compartment-specific, and the underlying mechanisms remain unknown.

Ubiquitously expressed aminoacyl–tRNA synthetases (ARSs) act as cross-over mediators of multiple biological processes for maintaining homeostasis, although they catalytically decode genetic information to yield the corresponding amino acids for protein synthesis[24,25]. In particular, the mammalian multi-tRNA synthetase complex (MSC), which comprises eight ARSs and three scaffold proteins (AIMP 1, 2, and 3), has emerged as central machinery involved in orchestrating a wide range of biological processes, including immunity, in a stimulus-dependent manner[26–32]. A representative component of the MSC is glutamyl-prolyl-tRNA synthetase 1 (EPRS1; eukaryotic ARSs are abbreviated using the single-letter code for their corresponding substrate amino acids, followed by "ARS1" for cytoplasmic enzymes and by "ARS2" for mitochondrial counterparts), which is a bifunctional enzyme comprising EARS1 fused with PARS1 via a linker containing a triple repeat of noncatalytic WHEP domains[28]. EPRS1 is phosphorylated at Ser886, Ser990, and Ser999, and these modifications activate multicellular functions in response to various stimuli, including IFNγ-induced translation-inhibition of inflammation-related mRNAs and RNA virus-specific antiviral immune regulation[26,30,32].

Although ARSs are essential ubiquitously expressed proteins involved in maintaining homeostasis, little is understood about their roles against inflammation. Given that the MSC is an immuno-surveillance system[26,30,32], we hypothesized that it regulates

inflammatory responses. We paid particular attention to EPRS1 because of its multiple functions in immunity. In this study, we evaluated the physiological roles of EPRS1 under inflammatory conditions in the context of myeloid cell-specific EPRS1-deficient mice. We found that EPRS1 specifically responds to inflammatory signals mediated by the TLR/PI3K/AKT axis and traffics AKT to early endosomes, where EPRS1 coordinates anti-inflammatory–specific AKT signaling for inflammation resolution.

## Results

### TLR-mediated inflammatory signaling triggers EPRS1 phosphorylation to promote anti-inflammatory immune regulation

Human MSC is multifaceted machinery, not only for protein synthesis but also for the control of physiological homeostasis. EPRS1, which resides exclusively in the MSC, acts as a molecular switch that triggers multiple cellular functions, including immune and metabolic regulation, via stimulus-dependent changes in its phosphorylation code[26,27,30]. To investigate the physiological role of EPRS1 in inflammation, we used heterozygous $Eprs1^{+/-}$ mice[30] in which EPRS1 mRNA and protein expression levels were significantly reduced in bone marrow-derived macrophages (BMDMs) (Fig. 1a, b). We examined the significance of EPRS1 in inflammatory regulation by comparing the wild-type (WT) $Eprs1^{+/+}$ and heterozygous $Eprs1^{+/-}$ mice. When BMDMs isolated from WT or heterozygous mice were stimulated with ligands of TLR1/2 (bacteria lipoprotein, BLP), TLR2/6 (Gram+ bacterial cell wall, zymosan), TLR4 (lipopolysaccharide, LPS), or TLR9 (oligodeoxynucleotides, CpG), reduced expression of EPRS1 by EPRS1 heterozygous cells increased secretion of inflammatory cytokines (TNF-α, IL-6, and MCP-1) significantly and decreased levels of the anti-inflammatory cytokine IL-10 (Fig. 1c–f). EPRS1, however, did not modulate cytokine production induced by poly(I:C) (a TLR3 stimulus) or IFNγ (Fig. 1c–f). We also investigated the role of the EPRS1-targeted signaling pathway by analyzing the TLR-activated signaling cascades using a luciferase promoter assay. As EPRS1-FLAG plasmid expression increased, EPRS1 inhibited TLR2-, TLR4-, and MYD88-induced activation of an NF-κB promoter-reporter in a dose-dependent manner (Supplementary Fig. 1a–c), suggesting that EPRS1 is involved in anti-inflammatory immune regulation.

Post-translational modifications of EPRS1, specifically phosphorylation at residues Ser886, Ser990, and Ser999, are key drivers of its dissociation from MSC for unique cell regulatory functions[26,27,30]. Hence, we evaluated the dependence of anti-inflammatory regulation on the modification of EPRS1 using specific anti-phospho-EPRS1 antibodies. The results revealed that residue Ser999, but not Ser990, is phosphorylated upon LPS stimulation in RAW 264.7 cells (Supplementary Fig. 1d). Notably, treatment of U937 cells with TLR2, TLR4, and TLR9 ligands induced phosphorylation of EPRS1 at Ser999, whereas treatment with the TLR3 ligand poly(I:C) did not affect this modification (Fig. 1g); the pattern of phosphorylation correlated with EPRS1-mediated immune responses to inflammatory stimulation (Fig. 1c–f). Because phosphorylation at Ser886 was constitutively detected in U937 cells even under unstimulated conditions (Fig. 1g), and the corresponding residue in mouse EPRS1 is asparagine, we further investigated phosphorylation at Ser999 under inflammatory conditions. Phosphorylation increased gradually from 15 to 60 min and persisted for 4 h, albeit with a slight decrease over this time period (Fig. 1h). A concentration of 10 ng/ml LPS was sufficient to phosphorylate EPRS1 on Ser999 (Supplementary Fig. 1e). Stimulation of RAW 264.7 or U937 cells with TLR ligands did not induce production of IFNγ (Supplementary Fig. 1f, g), indicating that TLR ligand-induced phosphorylation of Ser999 is independent of IFNγ (which also induces EPRS1 Ser999 phosphorylation), thereby initiating assembly of the IFNγ-activated inhibitor of translation (GAIT) complex that suppresses translation of inflammation-related mRNAs[26,32].

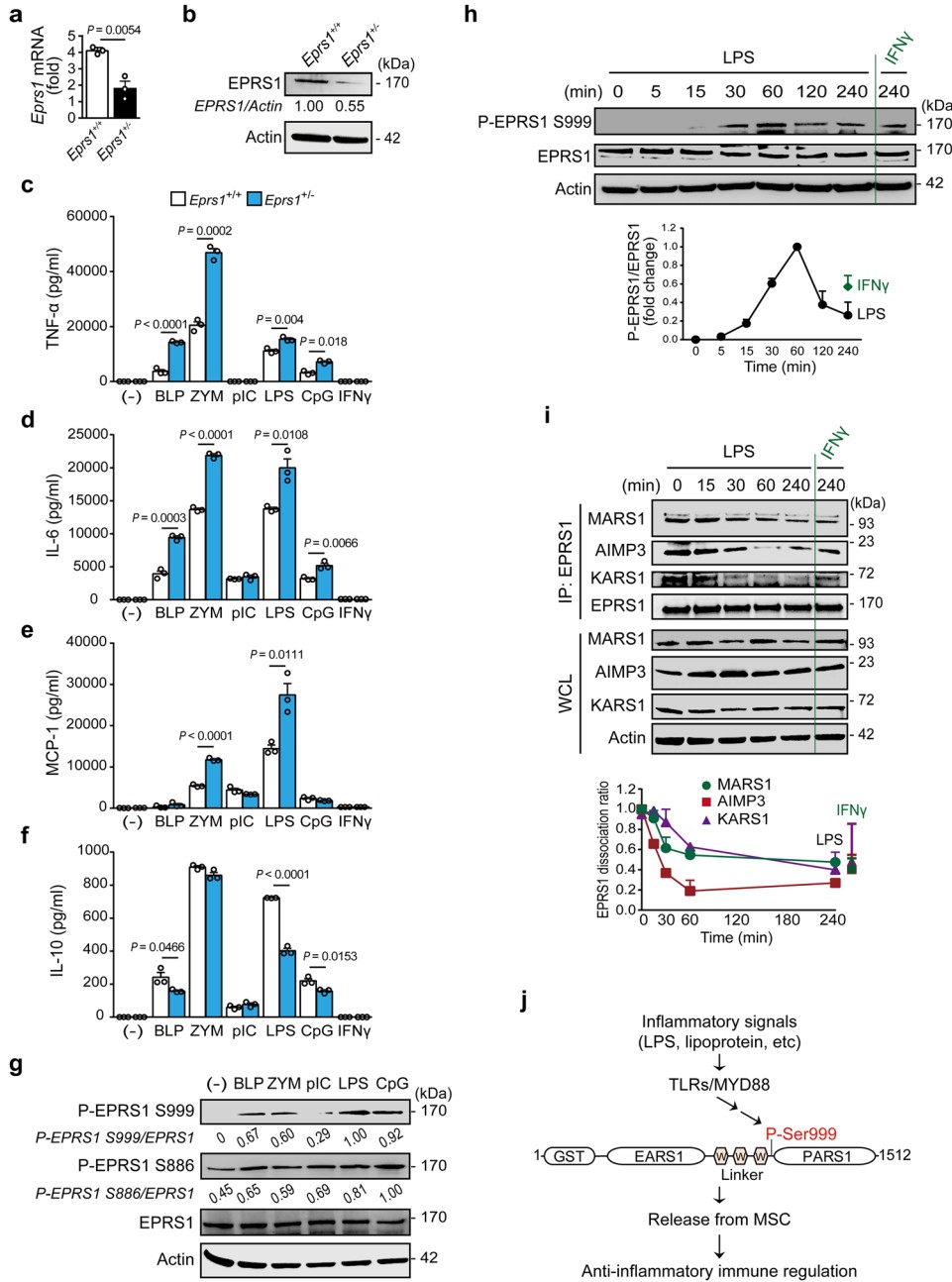

**Fig. 1 | TLR-mediated inflammatory signaling triggers EPRS1 phosphorylation and subsequent release from the MSC to negatively regulate inflammation. a, b** A comparison of the levels of *Eprs1* mRNA (**a**) and EPRS1 protein (**b**) in heterozygous *Eprs1*$^{+/-}$ mouse-derived BMDMs and WT BMDMs. The ratio of EPRS1 to actin is shown. **c–f** Enzyme-linked immunosorbent assay of the levels of the cytokines TNF-α (**c**), IL-6 (**d**), MCP-1 (**e**), and IL-10 (**f**) in supernatants from BMDMs isolated from *Eprs1*$^{+/+}$ and *Eprs1*$^{+/-}$ mice stimulated with TLR ligands or IFNγ (100 ng/ml) for 24 h. Bacterial lipoprotein Pam3CSK4 (BLP, 100 ng/ml), zymosan (ZYM, 10 μg/ml), poly(I:C) (pIC, 40 μg/ml), LPS (100 ng/ml), unmethylated cytosine-phosphate-guanine dinucleotides (CpG, 1 μg/ml), and IFNγ (100 ng/ml) were used. **g** Immunoblot analysis of EPRS1 phosphorylated at Ser999 and Ser886 in U937 cells treated for 1 h with BLP (100 ng/ml), ZYM (10 μg/ml), pIC (40 μg/ml), LPS (100 ng/ml), and CpG (1 μg/ml). The ratio of phosphorylated/total EPRS1 (p-EPRS1/EPRS1) is

shown. **h** Immunoblot analysis of phosphorylated EPRS1 in U937 cells following stimulation with LPS (100 ng/ml) for the indicated periods. IFNγ (100 ng/ml) was used as a positive control. Fold changes in the activation of EPRS1 are shown in the graph. **i** Endogenous immunoprecipitation (IP) with anti-EPRS1, followed by immunoblot analysis with anti-MARS1, anti-AIMP3, and anti-KARS1 antibodies at various times after treatment of U937 cells with LPS (100 ng/ml). IFNγ (100 ng/ml) was used as a positive control. The ratio of EPRS1 dissociation from MSC components was quantified based on the intensity of each protein band. **j** Schematic showing EPRS1 Ser999 phosphorylation induced by inflammatory stimuli. W WHEP. Data shown are representative of three independent experiments, each with similar results. All data are expressed as the mean ± SEM and were analyzed using a two-tailed unpaired *t*-test. Source data are provided as a Source Data file.

An immunoassay with antibodies against other MSC components (anti-MARS1, anti-AIMP3, and anti-KARS1) revealed that the interaction between EPRS1 and MSC components was reduced significantly following LPS stimulation, which is consistent with phosphorylation of EPRS1 at Ser999 (Fig. 1h, i). Note that human EPRS1, MARS1, AIMP2, and

AIMP3 are GST-like domain-containing proteins that tightly interact with each other via these domains within MSC[33]. Collectively, these results suggest that TLR/MYD88-mediated inflammatory signaling triggers EPRS1 phosphorylation at Ser999, freeing EPRS1 from the MSC to engage in anti-inflammatory regulation (Fig. 1j).

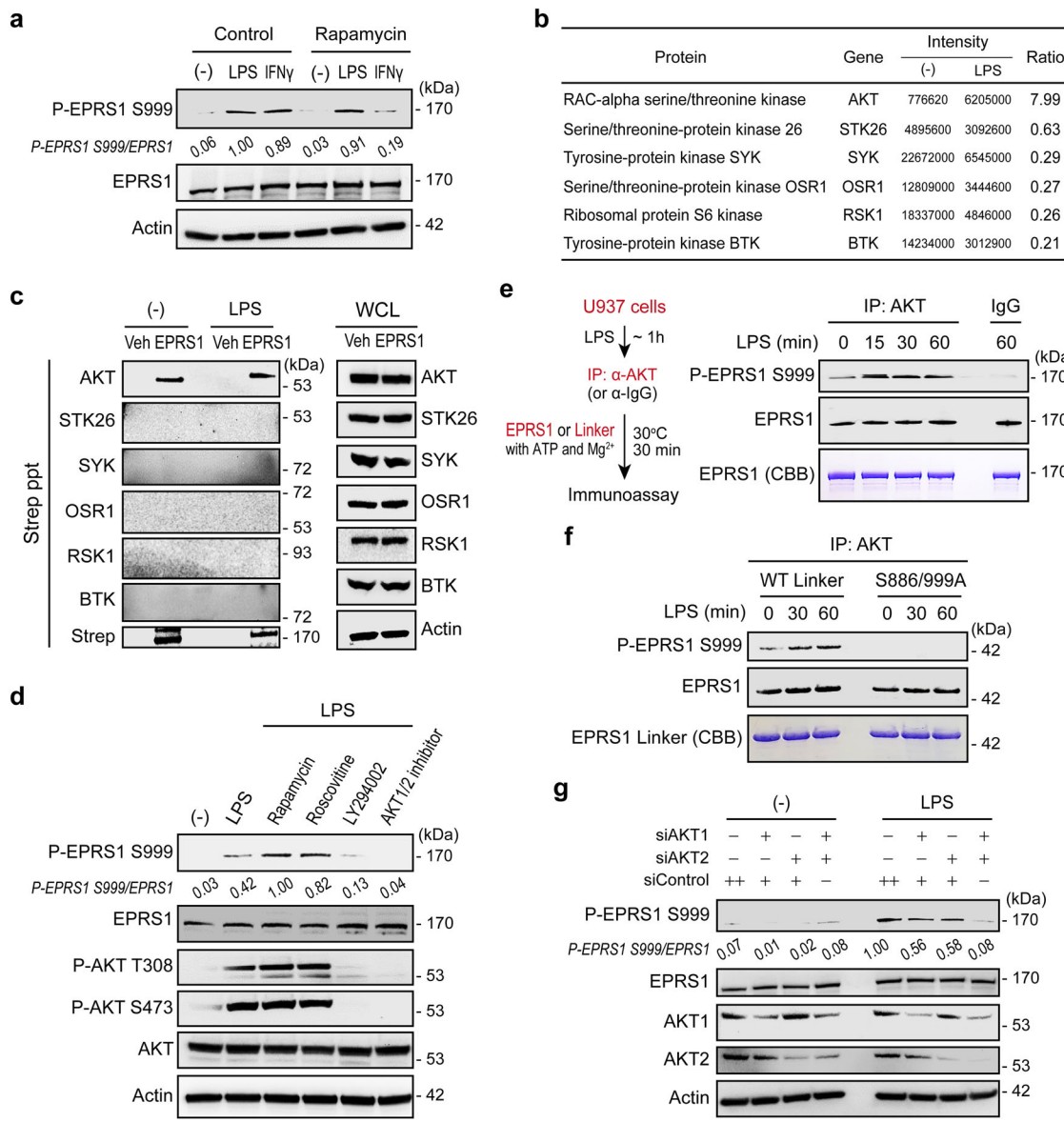

**Fig. 2 | Inflammatory-activated AKT phosphorylates EPRS1 at Ser999.**
**a** Immunoblot analysis of EPRS1 Ser999 phosphorylation in U937 cells treated for 4 h with LPS (100 ng/ml) or IFNγ (100 ng/ml). For pharmacological inhibition, cells were either untreated (control) or pretreated with rapamycin (10 nM) 30 min before stimulation with LPS or IFNγ. The ratio of phosphorylated/total EPRS1 is shown as p-EPRS1/EPRS1. **b** Proteomic analysis identified candidate kinases that bind preferentially to EPRS1. Protein intensity ratios under unstimulated (−)/stimulated (LPS) conditions are shown. **c** U937 cells, unstimulated (−) or stimulated with LPS (100 ng/ml) for 1 h, were lysed and incubated with vehicle (Veh) or purified Strep-EPRS1. The incubated cells were pulled down with Strep-Tactin beads for 1 h, and precipitates of endogenous kinases were analyzed by immunoblotting with the indicated antibodies. **d** U937 cells were pretreated with rapamycin (10 nM), roscovitine (10 μM), LY294002 (5 μM), or an AKT1/2 inhibitor (5 μM) for 30 min before LPS (100 ng/ml) stimulation. After 1 h, cells were harvested and the lysates were analyzed by immunoblotting with the indicated antibodies. The calculated ratio of

p-EPRS1/EPRS1 is shown. **e**, **f** Schematic representation of the AKT kinase assay protocol (**e**; left). An AKT-specific antibody was used to immunoprecipitate AKT from U937 cell lysates at the indicated times. The AKT-specific antibody bound to protein A/G-agarose beads was incubated with recombinant EPRS1 as a substrate for the in vitro kinase assay. AKT-mediated phosphorylation of Ser999 was determined using the EPRS1 full-length protein (**e**; right), or a recombinant Linker of wild-type (WT) or phosphorylation-resistant mutant (S886A/999A) EPRS1 (**f**), followed by immunoblotting with anti-EPRS1 Ser999. Rabbit IgG was used as a negative control for the AKT antibody (**e**). The purified EPRS1 proteins were subjected to SDS-PAGE and stained with Coomassie Brilliant Blue (CBB). **g** Effect of AKTs on Ser999 phosphorylation of EPRS1 in AKT1- and/or AKT2-suppressed and control RAW 264.7 cells treated with LPS (100 ng/ml) for 1 h. The fold change in activation of EPRS1 versus total EPRS1 is shown. All data are representative of three independent experiments, each with similar results. Source data are provided as a Source Data file.

## AKT activates the immune regulatory function of EPRS1 via phosphorylation at Ser999

We explored upstream kinases responsible for inflammatory signal-dependent phosphorylation of EPRS1 at Ser999. We first checked whether the upstream kinases responsible for IFNγ-mediated EPRS1 phosphorylation at Ser999 are also involved in inflammatory signal-induced EPRS1 phosphorylation. S6K1 is a key proximal kinase that

phosphorylates EPRS1 at Ser999 in IFNγ-stimulated BMDMs and insulin-treated adipocytes[27]. Rapamycin (mTORC1 inhibitor) inhibits S6K1 phosphorylation and suppresses the phosphorylation of EPRS1 at Ser999[34]. It was clear that treatment with rapamycin before LPS stimulation did not inhibit Ser999 phosphorylation, although it inhibited phosphorylation in IFNγ-stimulated U937 cells significantly (Fig. 2a), indicating that inflammatory signal-induced EPRS1 modification at

Ser999 is distinct from IFNγ or insulin-stimulated phosphorylation at the same residue.

To obtain mechanistic insight, we performed affinity purification mass spectrometry to identify cellular factors that associate with EPRS1 (Supplementary Fig. 2a). Of 721 proteins identified as proteins that potentially associate with EPRS1, we focused on kinases that may phosphorylate EPRS1 proximately (Fig. 2b and Supplementary Table 1). To select the kinases that specifically activate EPRS1 under inflammatory conditions, the intensity ratio of no stimulation to LPS stimulation was measured. Among six candidate kinases, AKT showed the highest intensity ratio, suggesting that it interacts with EPRS1 upon inflammatory stimulus (Fig. 2b). Interaction between the candidate kinases and EPRS1 was investigated further by incubating the purified Strep-EPRS1 with lysates from LPS-treated U937 monocytes, followed by detection with respective antibodies. A pull-down assay clearly showed that EPRS1 interacts with AKT, while no precipitation was observed with five other candidate kinases (Fig. 2c). In addition, we evaluated the interaction between EPRS1 and other key kinases precipitated from LPS-treated cells and found that EPRS1 interacted selectively with endogenous AKT (Supplementary Fig. 2b).

Next, we used kinase inhibitors to validate the proximal kinase activity. We observed significant inhibition of LPS-induced EPRS1 Ser999 phosphorylation following treatment with the PI3K inhibitor LY294002 (Supplementary Fig. 3a). MAPK inhibitors did not block EPRS1 phosphorylation (Supplementary Fig. 3b). Although Cdk5 plays an important role in initiating EPRS1 phosphorylation upon IFNγ treatment[26], the Cdk5 inhibitor (roscovitine) did not inhibit phosphorylation of EPRS1 at Ser999 in response to LPS stimulation (Supplementary Fig. 3c), suggesting that Cdk5 is not an upstream kinase necessary for EPRS1 phosphorylation, at least under inflammatory conditions. Based on the finding that EPRS1 interacts selectively with endogenous AKT (Fig. 2c), we subsequently tested other kinase inhibitors, including triciribine, an inhibitor of the AKT signaling pathway, and an AKT1/2 kinase inhibitor; AKT inhibition blocked phosphorylation at residues Thr308 and Ser473 and prevented phosphorylation of EPRS1 at Ser999 (Fig. 2d and Supplementary Fig. 3c). Following treatment with multiple kinase inhibitors, LPS-induced EPRS1 phosphorylation of Ser999 correlated well with phosphorylation of AKT at Thr308 and Ser473 (Fig. 2d and Supplementary Fig. 3c).

To confirm the phosphorylation of EPRS1 by AKT, we performed in vitro kinase assays using phosphorylated AKT immunoprecipitated from lysates of LPS-stimulated U937 cells (Fig. 2e). AKT activated by LPS stimulation phosphorylated the purified full-length EPRS1 at the indicated times (Fig. 2e). Moreover, LPS-activated AKT phosphorylated the WT EPRS1 linker domain (Linker), which joins the two catalytic domains EARS1and PARS1, but not a mutant Linker (S886A/S999A) (Fig. 2f), demonstrating that AKT is the proximal kinase responsible for the inflammation signal-specific EPRS1 modification. Furthermore, treatment with AKT1- and AKT2-specific siRNAs decreased EPRS1 phosphorylation at Ser999 in RAW 264.7 cells (Fig. 2g). A reduction in the amount of a single AKT isoform led to only slight inhibition of EPRS1 phosphorylation, suggesting that AKT1 and AKT2 can compensate for each other, at least in this experiment (Fig. 2g). Taken together, these data indicate that PI3K/AKT pathway inhibition impairs EPRS1 Ser999 phosphorylation.

Further investigation of the interaction between EPRS1 and AKT revealed that exogenously expressed EPRS1 and AKT1 proteins were capable of interacting with each other (Fig. 3a). To determine which region of EPRS1 was responsible for the interaction with AKT, we generated plasmid constructs encoding the GST-like, EARS1, Linker, and PARS1 domains. Strep pull-down assays revealed that the PARS1 domain was critical for AKT1 binding, with a smaller contribution from the EARS1 domain (Fig. 3b). The PARS1 domain is adjacent to Ser999 located in the C-terminal region of Linker, indicating that PARS1

provides a docking site for AKT that directs the active site of the kinase proximate to the target site in the EPRS1 Linker.

We next analyzed the AKT domains involved in this interaction. Co-immunoprecipitation of individual AKT1 domains and EPRS1 revealed that the AKT1 kinase domain was essential for the EPRS1 interaction (Fig. 3c, d). Concomitantly, the formation of the signaling complex by EPRS1 did not show specificity for AKT isoforms. Rather than exerting specific control over AKT types, EPRS1 interacted with AKT1, AKT2, and AKT3 (Fig. 3e). Because all AKT isoforms can phosphorylate GSK3β at Ser9[35], the EPRS1-mediated anti-inflammatory effects with all isoforms may occur via similar mechanisms.

However, the endogenous interaction between EPRS1 and AKT in U937 cells was dependent on inflammatory stimulation (LPS). The EPRS1–AKT interaction strengthened over time, peaked at 30 min, and was associated with GSK3β in LPS-stimulated U937 cells (Fig. 3f). No direct interaction between EPRS1 and GSK3β was observed, suggesting that the endogenous EPRS1/GSK3β complex is mediated by AKT (Supplementary Fig. 3d). In addition, confocal microscopy following 30 min of LPS stimulation revealed that in HeLa cells, EPRS1 was driven to distinct cellular locations to colocalize with AKT (Fig. 3g).

The PI3K/AKT pathway plays an essential role in preserving the integrity of the immune system[10,36], and AKT is a key signaling protein that controls the production of inflammatory cytokines[37]. Taken together, our data suggest that the TLR-activated PI3K/AKT axis phosphorylates EPRS1 at Ser999 and converts it to an anti-inflammatory immune regulator via association with AKT and, potentially, GSK3β (Fig. 3h).

## EPRS1 is an effector protein that coordinates early endosomal anti-inflammatory AKT signaling

Our results revealed that EPRS1 modification is controlled by the PI3K/AKT signaling pathway, whose signaling dynamics are regulated at the plasma membrane and endosomes[38–40]. To determine the physiological role of EPRS1, we evaluated the cellular localization of EPRS1 using purified recombinant full-length EPRS1. EPRS1 was overlaid onto phospholipid-coated strips, and protein binding was assessed. EPRS1 clearly bound to monophosphate inositol lipids, with especially high specificity for the predominant lipid, PI(3)P, on early endosomal membranes[41]; by contrast, binding to PI(4)P was relatively weak (Fig. 4a).

Interestingly, amino acid sequence analysis revealed that the EPRS1 Linker contains a conserved sequence responsible for binding to PI(3)P-positive endosomes (Fig. 4b) found in the PTEN CBR3 loop[23]. Mutation of this consensus sequence abolished the interaction with PI(3)P entirely (Fig. 4c). PI(3)P plays a crucial role in recruiting effector proteins to early endosomes[42]. Moreover, the high-intensity ratio of EPRS1-associated AKT upon LPS treatment suggests dynamic changes in intracellular localization of AKT, which activate EPRS1 in addition to the dissociation of EPRS1 from the MSC (Fig. 2b). Therefore, we assessed the cellular localization of proteins identified from the proteome (Supplementary Table 1) based on the protein annotation through evolutionary relationships (PANTHER) classification system[43], i.e., any process in which a protein is transported to a specific location at the level of a cell. The analysis revealed that EPRS1 may associate with Rab5B and Rab5C and then form a complex with AKT1 (Supplementary Fig. 4a). Indeed, the pull-down assay using EPRS1-GFP demonstrated that EPRS1 interacted with Rab5 and that the EARS1 domain was responsible for the interaction with Rab5 (Supplementary Fig. 4b).

Endosomal membranes serve as a physical platform for discriminating signaling complex assemblies[44,45]. The presence of the GTP-bound activated form of Rab5 with PI(3)P at early endosomal membranes acts as a signal for the recruitment of effector proteins[46–48]. Hence, we sought to determine whether EPRS1 is

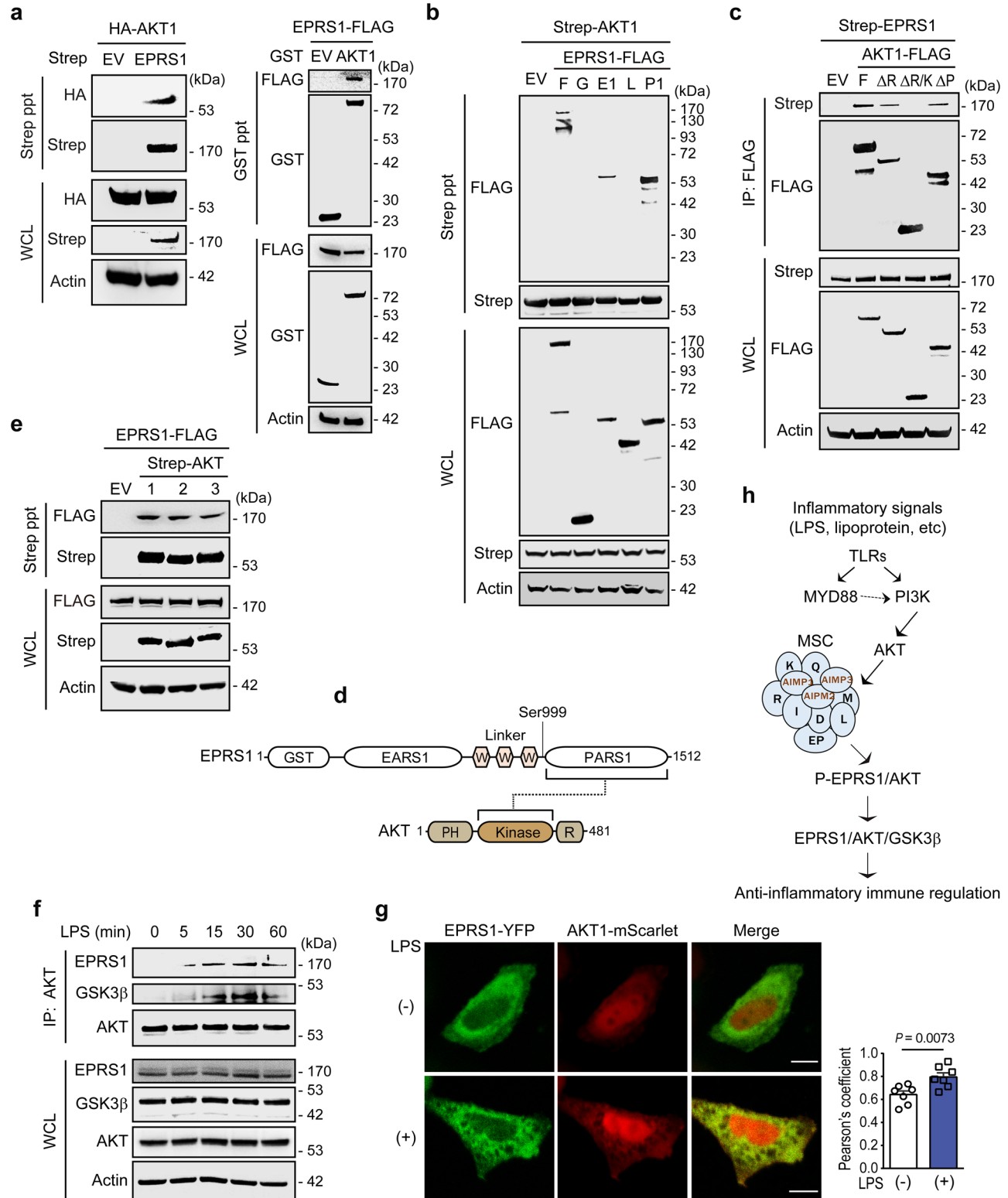

associated with Rab5 at early endosomes. Cells co-transfected with EPRS1-YFP and Rab5-mCherry exhibited colocalization (Fig. 4d) upon LPS stimulation. However, EPRS1 did not colocalize with other organelle markers, including Rab7 (late endosome), LAMP-1 (lysosome), and LC3 (autophagosome), in HeLa cells (Fig. 4d). To confirm localization of EPRS1 to Rab5-positive vesicles, we used an activation-mimic form of Rab5(Q79L)-CFP along with EPRS1-YFP and then stained activated AKT with anti-AKT Ser473 antibody to trace their location to the early endosomal membrane upon LPS stimulation (Fig. 4e and

Supplementary Fig. 4c). Although EPRS1 plays a pivotal role in AKT translocation to Rab5-positive endosomes, the proteins were not colocalized when the lipid-binding sequence in the linker domain was mutated (Fig. 4e). Because EPRS1 is ubiquitous in cytoplasm, and most of it is still associated with MSC, we used the EPRS1 Linker-GFP to show clear colocalization with Rab5(Q79L)-mCherry (Fig. 4f and Supplementary Fig. 4d). The EPRS1 Linker constitutively associated with Rab5(Q79L)-positive endosomes, regardless of LPS stimulation (Fig. 4f), providing evidence that inflammatory signal-dependent

**Fig. 3 | EPRS1 interacts directly with AKT and associates with GSK3β.**
**a** Immunoassay of 293T cells cotransfected with Strep-tagged empty vector (EV) or EPRS1 with HA-AKT1 (left), and GST-tagged EV or AKT1 with EPRS1-FLAG (right), followed by Strep or Glutathione precipitation and immunoblotting with an anti-HA or an anti-FLAG antibody. **b** Immunoassay of 293T cells cotransfected with Strep-AKT1 and FLAG-tagged EPRS1 constructs, followed by Strep precipitation and immunoblotting with an anti-FLAG antibody. F full-length (aa 1–1512), G GST-like domain (aa 1–196), E1 EARS1 (aa 197–682), L Linker (aa 683–1023), P1 PARS1 (aa 1024–1512). **c** Immunoassay of the interaction between EPRS1 and AKT1 in lysates of 293T cells expressing Strep-EPRS1 and AKT1-FLAG proteins, assessed by anti-FLAG immunoprecipitation followed by immunoblot analysis with an anti-Strep antibody. F full-length (aa 1–481), ΔR deletion of regulatory domain (aa 1–409), ΔR/K deletion of regulatory/kinase domains (aa 1–152), ΔP deletion of PH domain (aa 153–481). **d** Schematic diagram showing the interaction between EPRS1 and AKT. **e** Immunoassay of EPRS1-FLAG co-expressed with Strep-AKT isoforms in 293T cells.

Strep-AKT isoforms were pulled down with Strep-Tactin beads and co-precipitation of EPRS1 was detected by immunoblotting with an anti-FLAG antibody. **f** Use of an anti-AKT antibody to immunoprecipitate endogenous AKT from U937 cells stimulated with LPS (100 ng/ml) at the indicated times. Co-IP of endogenous EPRS1 and GSK3β was assessed using anti-EPRS1 and anti-GSK3β antibodies, respectively. **g** Confocal microscopy analysis of colocalization of EPRS1 (green) and AKT1 (red) in transfected HeLa cells treated with LPS (500 ng/ml) for 30 min. Scale bars, 10 μm. The colocalization index of EPRS1 and AKT1 upon LPS stimulation was quantified using Pearson's correlation coefficient (right, $n = 7$ cells). Error bars indicate the mean ± SEM. Data were analyzed using Student's two-tailed $t$-test. **h** Schematic of signaling pathways responsible for EPRS1 Ser999 phosphorylation and the sequential interactions between downstream proteins involved in negative regulation of inflammation. D DARS1, EP EPRS1, I IARS1, K KARS1, L LARS1, M MARS1, Q QARS1, R RARS1. Data are representative of three independent experiments, each with similar results. Source data are provided as a Source Data file.

release of EPRS1 from the MSC is required for localization to early endosomes.

We then isolated early endosomes and confirmed that LPS-stimulated EPRS1 (i.e., phosphorylated EPRS1) translocated from the cytoplasm to early endosomes. Upon LPS stimulation, both AKT1 and AKT2 compartmentalized to early endosomes along with EPRS1 (Fig. 4g). Taken together, these results strongly support the concept that MSC-dissociated phospho-EPRS1 translocates to the early endosomal membrane to promote assembly of an endosome-specific anti-inflammatory AKT signaling complex: the PARS1 domain interacted with and trafficked AKT to early endosomes via the Linker lipid-binding region, and the EARS1 domain interacted with Rab5 (Fig. 4h).

### EPRS1 ameliorates inflammation in macrophages and mice

To investigate the early endosome-specific functions of EPRS1, we examined TLR4-mediated inflammatory downstream signaling pathways. To validate EPRS1-mediated anti-inflammatory signaling in macrophages, we deleted EPRS1 from the myeloid lineage by crossing $Eprs1^{fl/fl}$ with $Lyz2^{Cre}$ mice (hereafter referred to as $Eprs1^{fl/fl}Lyz2^{Cre}$). First, we confirmed the kinetics of EPRS1 Ser999 phosphorylation along with AKT activation upon LPS stimulation in BMDMs from WT ($Eprs1^{fl/fl}$) mice (Supplementary Fig. 5a). Relative to BMDMs from $Eprs1^{fl/fl}$ mice, EPRS1-deficient BMDMs from $Eprs1^{fl/fl}Lyz2^{Cre}$ mice exhibited poor AKT activation following LPS stimulation, and GSK3β Ser9 phosphorylation was reduced significantly (Fig. 5a). Phosphorylation of GSK3β at Ser9 inhibits its activity, which in turn triggers anti-inflammatory cytokine production and negatively regulates NF-kB-mediated signaling via CREB activation[49–52]. Thus, the lower GSK3β phosphorylation in EPRS1-deficient BMDMs decreased the phosphorylation of CREB while increasing the phosphorylation of NF-kB (Fig. 5a). However, phosphorylation of mTOR and S6K was unaffected in both WT and EPRS-deficient BMDMs following LPS stimulation (Supplementary Fig. 5b), providing further evidence that the consequences of inflammatory signal-induced EPRS1 phosphorylation at Ser999 are different from those of IFNγ or insulin-stimulated phosphorylation at the same residue[27,34]. At the same time, this implies that the function of EPRS1 under inflammatory conditions is confined to the early endosomal compartment. Note that we used a cell culture medium containing low levels of serum and did not change the medium during LPS stimulation; this was done to exclude the possibility of mTOR activation by any growth factors or nutrients. Moreover, AKT-mediated mTORC1 signaling does not occur at the level of early endosomes; rather, it is activated during lysosome translocation[20]. Phosphorylation of AKT was also reduced significantly in siEPRS1-transfected RAW 264.7 cells following LPS stimulation, and subsequent AKT-mediated GSK3β Ser9 phosphorylation was reduced markedly (Fig. 5b and Supplementary Fig. 5c). Thus, the data demonstrate that EPRS1 is an effector protein involved in assembling an anti-inflammatory complex in the early

endosome, which alleviates inflammation-associated downstream signaling.

To evaluate the roles of EPRS1 in anti-inflammatory regulation, we used myeloid-specific EPRS1 knockdown $Eprs1^{fl/fl}Lyz2^{Cre}$ mice. Under inflammatory stimulation, TNF-α and IL-6 were produced at significantly higher levels in BMDMs derived from $Eprs1^{fl/fl}Lyz2^{Cre}$ mice than in BMDMs derived from $Eprs1^{fl/fl}$ mice, whereas IL-10 levels were attenuated in $Eprs1^{fl/fl}Lyz2^{Cre}$ cells (Fig. 5c–e), which is consistent with the results obtained using heterozygous $Eprs1^{+/-}$ mice (Fig. 1c–f). Inflammatory stimulation (i.e., with LPS) of peritoneal macrophages isolated from $Eprs1^{fl/fl}Lyz2^{Cre}$ mice also increased production of TNF-α and IL-6 while decreasing IL-10 production (Fig. 5f–h). Similar results were obtained following the siRNA-mediated knockdown of EPRS1 in RAW 264.7 cells (Fig. 5i–k). Furthermore, we used protein arrays to analyze the production of chemokines and cytokines in LPS-stimulated BMDMs and peritoneal macrophages from $Eprs1^{fl/fl}$ and $Eprs1^{fl/fl}Lyz2^{Cre}$ mice. The results showed that EPRS1 negatively regulates the production of inflammatory cytokines (Supplementary Fig. 6).

To determine the functional importance of EPRS1 during inflammation in vivo, we used representative mouse models of inflammatory diseases[53]. We first challenged $Eprs1^{fl/fl}$ and $Eprs1^{fl/fl}Lyz2^{Cre}$ mice with LPS. EPRS1-deficient mice were more susceptible to LPS-induced septic shock (Fig. 6a). Consistent with the in vitro results, the concentration of IL-10 in serum was lower in $Eprs1^{fl/fl}Lyz2^{Cre}$ mice than in $Eprs1^{fl/fl}$ mice at 6 h post-LPS administration (Fig. 6b). By contrast, EPRS1-knockdown mice produced much higher levels of the pro-inflammatory cytokines IL-6 and MCP-1 in serum than WT mice at 24 h (Fig. 6c, d). The spleen weight was also higher in LPS-challenged $Eprs1^{fl/fl}Lyz2^{Cre}$ mice (Supplementary Fig. 7a). Further histological studies revealed that severe lung injuries, such as the disruption of alveolar walls and widespread alveolar wall thickening, were exacerbated in LPS-stimulated $Eprs1^{fl/fl}Lyz2^{Cre}$ mice (Fig. 6e). Moreover, infiltration of inflammatory cells into the lung, liver, and kidney was more extensive in LPS-treated $Eprs1^{fl/fl}Lyz2^{Cre}$ mice (Fig. 6e and Supplementary Fig. 7b, c).

Next, we infected $Eprs1^{fl/fl}$ and $Eprs1^{fl/fl}Lyz2^{Cre}$ mice with *Salmonella typhimurium*, a Gram-negative bacteria known to cause systemic cytokine production and sepsis[54]. Similar to the effect of the bacterial product LPS, EPRS1-deficient mice were more susceptible to septic shock induced by *S. typhimurium* infection (Fig. 6f). The level of the anti-inflammatory cytokine IL-10 in serum was reduced in $Eprs1^{fl/fl}Lyz2^{Cre}$ mice, whereas the levels of pro-inflammatory cytokines IL-6 and MCP1 were higher (Fig. 6g–i). Infiltration of inflammatory cells into the lung was more extensive in *S. typhimurium*-infected $Eprs1^{fl/fl}Lyz2^{Cre}$ mice (Fig. 6j).

We further examined chronic intestinal inflammation using a dextran sulfate sodium (DSS)-induced colitis model. Because colitis is a complicated inflammatory disease in which both adaptive and innate

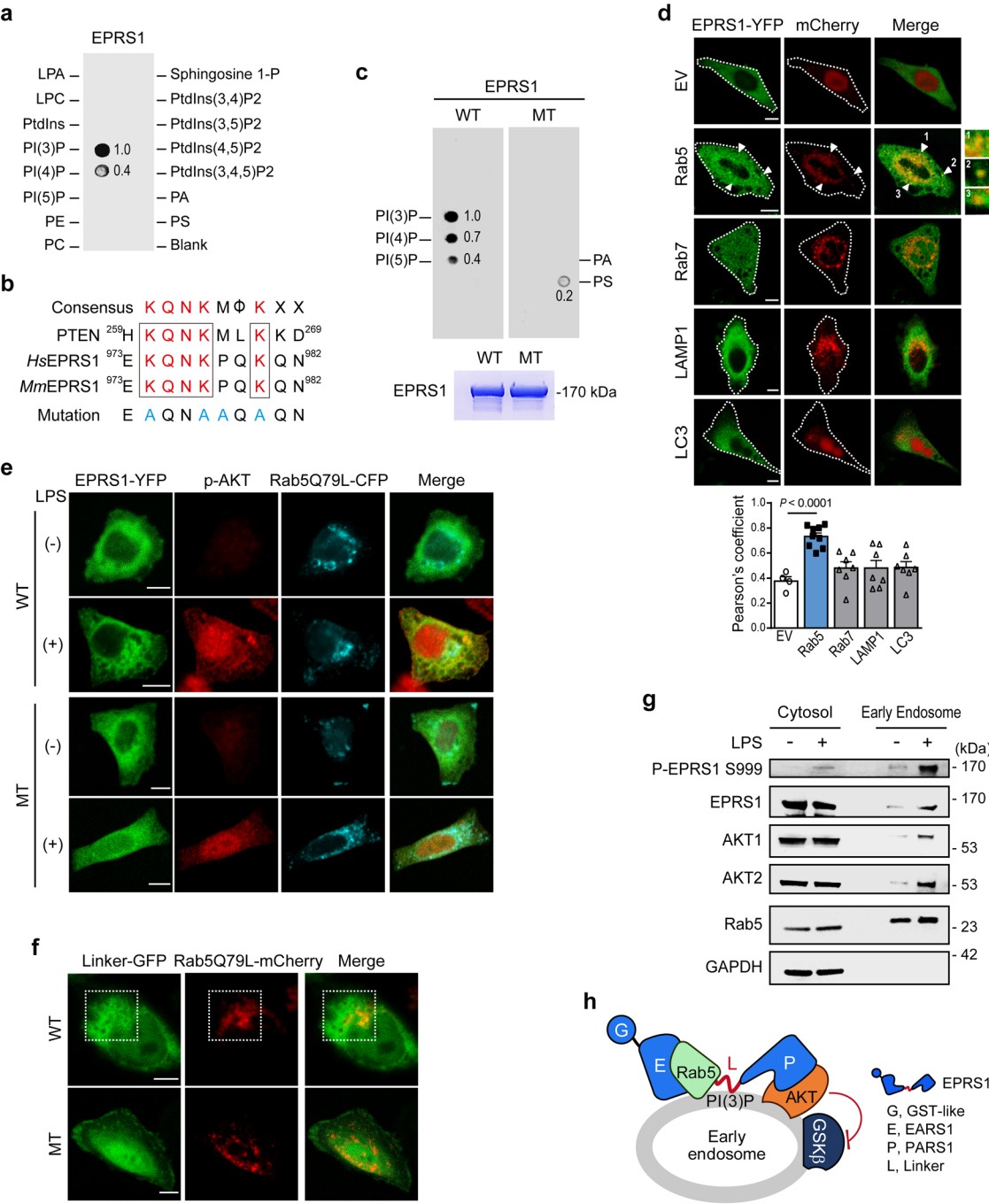

**Fig. 4 | EPRS1 binds specifically to PI(3)P and Rab5 to coordinate early endosomal anti-inflammatory AKT signaling. a** PIP strips precoated with indicated lipids were incubated with purified Strep-EPRS1 and probed with an anti-Strep antibody. **b** Sequence alignment of EPRS1 with PTEN to show the PI(3)P-binding consensus motif. The last row shows mutations engineered to neutralize early endosomal localization. **c** PIP strips were incubated with purified Strep-EPRS1 WT or its mutant (MT) and probed with an anti-Strep antibody for the same exposure time. **d** HeLa cells were transfected with a mCherry-tagged EV, Rab5, Rab7, LAMP1, or LC3 (red), in combination with EPRS1-YFP (green). Confocal microscopy analysis showing colocalization of EPRS1 and organelle-specific proteins in transfected cells following LPS (500 ng/ml) stimulation for 30 min. White arrowheads indicate colocalized early endosomes, and the magnified images are displayed in the right panel. The colocalization index of EPRS1 and specific organelle markers was quantified using Pearson's correlation (bottom, $n = 5$ cells for EV, $n = 9$ cells for Rab5, $n = 7$ cells for other markers). Error bars indicate the mean ± SEM. Data were analyzed using Student's two-tailed $t$-test. **e** HeLa cells were transfected with YFP-

tagged EPRS1 (green) WT or non-endosomal associated MT in combination with Rab5(Q79L)-CFP (cyan). After 16 h, cells were fixed following LPS stimulation for 30 min and then analyzed by immunofluorescence with an antibody specific for p-AKT (red). Scale bars, 10 μm. **f** Confocal microscopy analysis of colocalization of EPRS1 Linker-GFP (green) and Rab5(Q79L)-mCherry (red) in live HeLa cells. MT, K974A/K977A/P978A/K980A mutant. Colocalized images in the white dotted boxes. Scale bars, 10 μm. **g** LPS (100 ng/ml) treatment of U937 cells induces translocation of Ser999-phosphorylated EPRS1 and AKTs to Rab5-positive early endosomes. An anti-GAPDH antibody was used as a cytosolic marker. **h** Schematic depicting the interaction between EPRS1 and AKT in the Rab5-positive early endosome to regulate downstream GSK3β signaling. The conserved region within the linker domain (aa 974–980), located in the spacer between the third WHEP repeat domain and the PARS1 domain, is important for PI(3)P binding. Data are representative of three independent experiments, each with similar results. Source data are provided as a Source Data file.

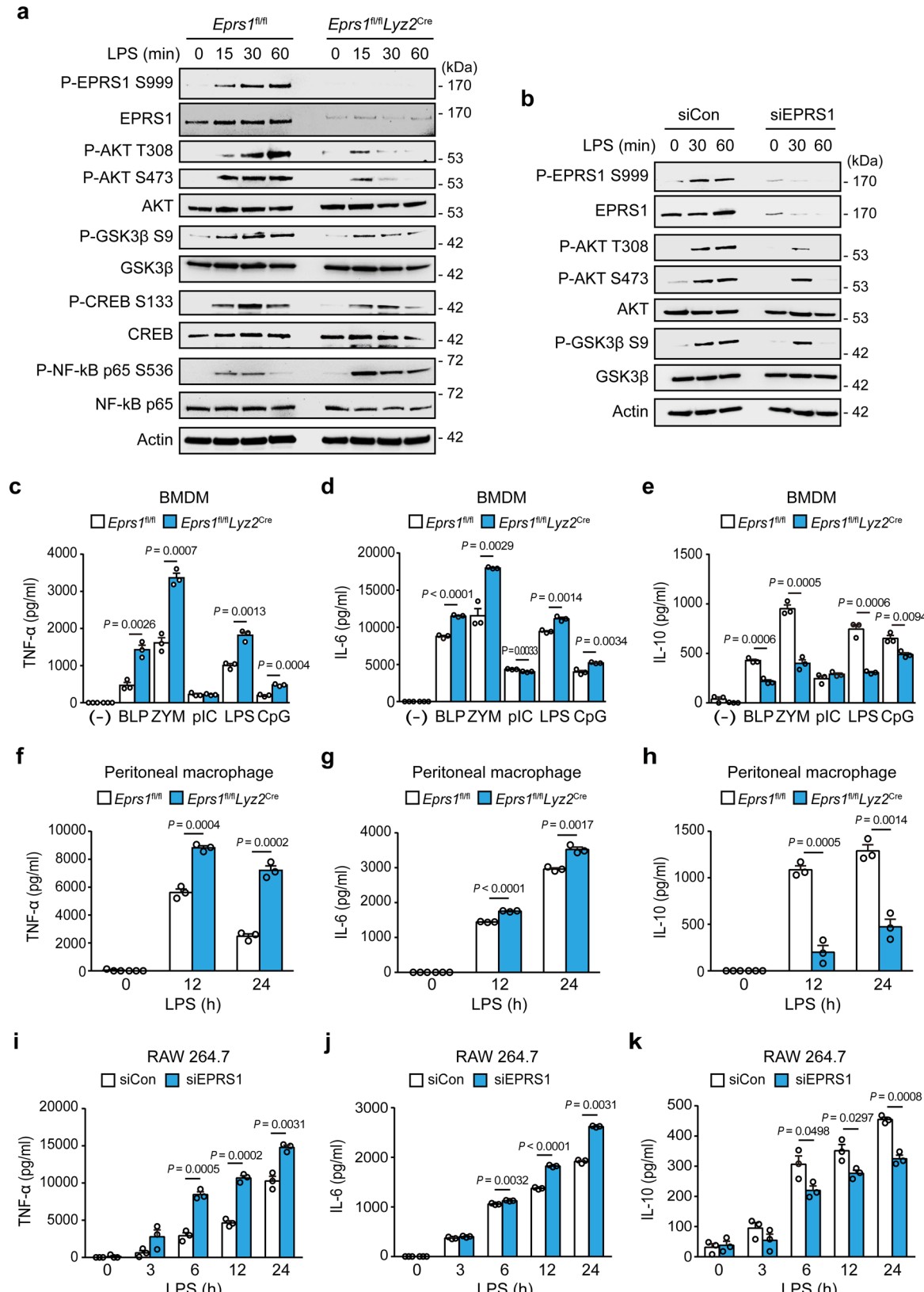

immune cells are involved[55], we first monitored the anti-inflammatory effects of EPRS1 in heterozygous *Eprs1*[+/−] mice. The results showed that EPRS1 whole-body knockdown (*Eprs1*[+/−]) mice were more sensitive to DSS-induced colitis than WT mice (Supplementary Fig. 8a–e). During the acute phase of colitis, innate immunity, which is dominated by macrophages, plays a critical role in the pathogenesis of colitis[56,57].

Therefore, we used myeloid-specific EPRS1 knockdown *Eprs1*[fl/fl]*Lyz2*[Cre] mice. Body weight loss and disease activity scores were higher in *Eprs1*[fl/fl]*Lyz2*[Cre] colitis mice than in *Eprs1*[fl/fl] colitis mice from Day 6 after DSS treatment (Fig. 6k, l). In addition, the colon length was significantly shorter in *Eprs1*[fl/fl]*Lyz2*[Cre] mice (Fig. 6m). Histological analyses revealed pronounced submucosal inflammatory cell infiltration of the colon in

**Fig. 5 | EPRS1 negatively regulates inflammation in TLR-stimulated macrophages. a** Immunoblot analysis of phosphorylated- and total EPRS1, AKT, GSK3β, CREB, and NF-kB p65, and of actin, in LPS-treated BMDMs isolated from *Eprs1*[fl/fl] and *Eprs1*[fl/fl]*Lyz2*[Cre] mice. **b** Immunoblot analysis of cell lysates from LPS-treated RAW 264.7 cells transfected with EPRS1-specific siRNA (siEPRS1) or control (non-targeting) siRNA (siCon) using the indicated antibodies. **c–e** Enzyme-linked immunosorbent assay of TNF-α (**c**), IL-6 (**d**), and IL-10 (**e**) in supernatants from BMDMs isolated from *Eprs1*[fl/fl] and *Eprs1*[fl/fl]*Lyz2*[Cre] mice stimulated for 24 h with TLR ligands (BLP, 100 ng/ml; ZYM, 10 μg/ml; poly(I:C), 40 μg/ml; LPS, 100 ng/ml, and CpG, 1 μg/ml). **f–h** Concentration of TNF-α (**f**), IL-6 (**g**), and IL-10 (**h**) in supernatants from peritoneal macrophages isolated from *Eprs1*[fl/fl] and *Eprs1*[fl/fl]*Lyz2*[Cre] mice stimulated with LPS (100 ng/ml) for 12 or 24 h, as quantified by ELISA. **i–k** Concentrations of TNF-α (**i**), IL-6 (**j**), and IL-10 (**k**) in supernatants from RAW 264.7 cells transfected with siCon or siEPRS1, followed by stimulation with LPS (100 ng/ml) for the indicated times. The data shown are representative of three independent experiments, each with similar results. Data are expressed as the mean ± SEM. Two-tailed unpaired *t*-tests were used. Source data are provided as a Source Data file.

*Eprs1*[fl/fl]*Lyz2*[Cre] mice, and disruption of the epithelial lining and the damage score (particularly in the distal region) were high (Fig. 6n, o and Supplementary Fig. 8f).

Collectively, these results strongly indicate that EPRS1 is a critical effector protein that helps maintain physiological homeostasis during inflammation via the following series of events. TLR-mediated inflammatory signal-activated PI3K/AKT axis phosphorylates EPRS1 at Ser999, which releases EPRS1 from the MSC. The released EPRS1 is converted to an anti-inflammatory immune regulator via trafficking AKT into the early endosomal membrane to assemble an endosome-specific AKT signaling complex in association with Rab5. These events subsequently promote AKT activation-mediated GSK3β phosphorylation, which increases anti-inflammatory cytokine production, leading to inflammation resolution (Fig. 6p).

## Discussion

Inflammation is essential for efficient immunity, healing, and returning to homeostasis following pathogen invasion or harmful endogenous signals. Upon inflammatory stimulation, TLR-activated PI3K converts phosphatidylinositol-(4,5)-bisphosphate (PIP2) to phosphatidylinositol-(3,4,5)-trisphosphate (PIP3), which in turn activates AKT at the plasma membrane[58]. Once activated, AKT freely redistributes to other subcellular compartments, where it performs specific functions[59–63]. Compartment-specific signaling allows the cell to respond to external stimuli, coordinate signaling cascades in time and space, and maintain homeostasis; however, its disturbance can cause deleterious effects, including immunological disorders[20,64]. The mechanisms underlying organelle-specific AKT signaling, particularly that involved in regulating inflammation, remain largely unknown. Here, we showed that AKT activated by the TLR/PI3K axis specifically targets MSC-associated EPRS1 and modifies it at Ser999. Ser999 is located in the linker domain, which is on the outside of the MSC[28,33,65] and is thus readily accessible to the kinase. EPRS1 released from the MSC following phosphorylation maintains full activation of AKT by associating with PI(3)P- and Rab5-positive early endosomes, which eventually serves as a specific platform for AKT signaling, thereby ensuring the specificity of AKT signaling for anti-inflammatory regulation. We discovered that EPRS1 contains a consensus PI(3)P lipid-binding sequence that drives its early endosomal trafficking and compartmentalization of AKT into Rab5-positive early endosomes under inflammatory conditions sensed by TLRs (Fig. 4). This results in full AKT activation, inhibiting GSK3β to suppress inflammatory responses (Fig. 5), leading ultimately to resolution of inflammation (Fig. 6).

It is well-known that the cytoplasmic MSC serves as a reservoir for ARSs, which can respond rapidly to cellular stresses without making demands on de novo transcription and translation[30,66]. Although the fundamental function of the MSC is not well understood, it is known that in vivo depletion of the AIMP2 (previously named p38), the known scaffold protein required for the assembly of the MSC, does not affect global protein synthesis and cell growth[67]. Only a small fraction of such components are released from the MSC to conduct non-translational functions in a stimulus-dependent manner, and this release does not alter overall protein synthesis efficiency[68]. In this regard, a recent study demonstrated that there is no change in global protein synthesis and cell growth even in the absence of RARS1 and QARS1 in the MSC[69]. This study further investigated whether the exclusion of RARS1 and QARS1 from the MSC hinders mRNA translation. The results showed that the levels of newly synthesized proteins are well maintained in cells lacking MSC-localized RARS1 and QARS1[69]. EPRS1 released from the MSC preserves its translational role and does not disrupt total protein synthesis[27,30,70]. Based on these previous results, we suggest that the release of inflammatory signal-specific EPRS1 from the MSC would not reduce mRNA translation. Thus, it is unlikely that EPRS1-mediated anti-inflammatory immune activation is linked to a decrease in the translation of pro-inflammatory mediator genes.

The EPRS1 linker domain contains three serine residues that are phosphorylated in response to stimulation (Ser886, Ser990, and Ser999) and activate distinct signaling events through interactions with effector proteins. Both insulin and inflammatory signals induce phosphorylation of EPRS1 at Ser999, but EPRS1 released from MSC interacts with multiple interactors including ribosomal protein L13a and GAPDH[26,70], FATP1[27], or AKT (Fig. 3). Although the Ser999 phosphorylation site is shared, the various roles of EPRS1 can be attributed to distinct stimulus-dependent kinases. IFNγ and insulin activate Cdk5- and mTORC1-directed multisite phosphorylation of S6K1, which phosphorylates EPRS1 at Ser999; therefore, the mTOR inhibitor rapamycin critically blocks this phosphorylation[34]. However, we showed that rapamycin treatment under conditions of LPS stimulation still phosphorylated EPRS1 at Ser999 via AKT activation (Fig. 2d and Supplementary Fig. 3c), indicating that EPRS1 is a stimulus-dependent molecular switch that drives specific cell signaling pathways.

While EPRS1 integrates signals from multiple stimuli such as infection, inflammation, and metabolic disorders, there are apparent differences in the kinetics of EPRS1 phosphorylation and physiological function[26,27,30]. In the case of inflammatory conditions, EPRS1 seems to play a compensatory role to resolve inflammation by coordinating the AKT signaling complex (this study), as well as by assembling the GAIT complex[26]. Our results showed that inflammatory signal (LPS)-activated PI3K/AKT turns on EPRS1 Ser999 phosphorylation within 15 min, reaching a maximum phosphorylation level at around 1 h (Figs. 1h, 2e, and 5a). IFNγ-stimulated EPRS1 phosphorylation at Ser886 and Ser999 starts after 1 and 2 h of stimulation, respectively, and persists for up to 24 h[26]. Phosphorylation at both Ser886 and Ser999 is required for EPRS1 exit from the MSC and for the formation of the active GAIT complex that binds the GAIT element in the 3′UTR of inflammation-associated mRNAs and represses translation of inflammatory gene expression[26]. Of note, the translational silencing procedure takes about 14–16 h under conditions of prolonged IFNγ stimulation[26], which is far slower than PI3K/AKT-driven EPRS1 anti-inflammatory immune regulation. Prolonged expression of inflammatory genes contributes to the progression of various diseases, including cancer and Alzheimer's disease. EPRS1-associated GAIT-mediated translational control of inflammatory transcripts may serve to protect cells from inflammation and injury in the presence of persistent inflammatory stimuli[71]. Our results also showed that TLR ligands, including LPS, do not induce IFNγ production (at least not within 24 h) in RAW 264.7 or U937 cells (Supplementary Fig. 1f, g). These findings suggest that TLR/PI3K/AKT-activated EPRS1 phosphorylation at Ser999 directs the first line of anti-inflammatory immune defense against inflammatory signals, including

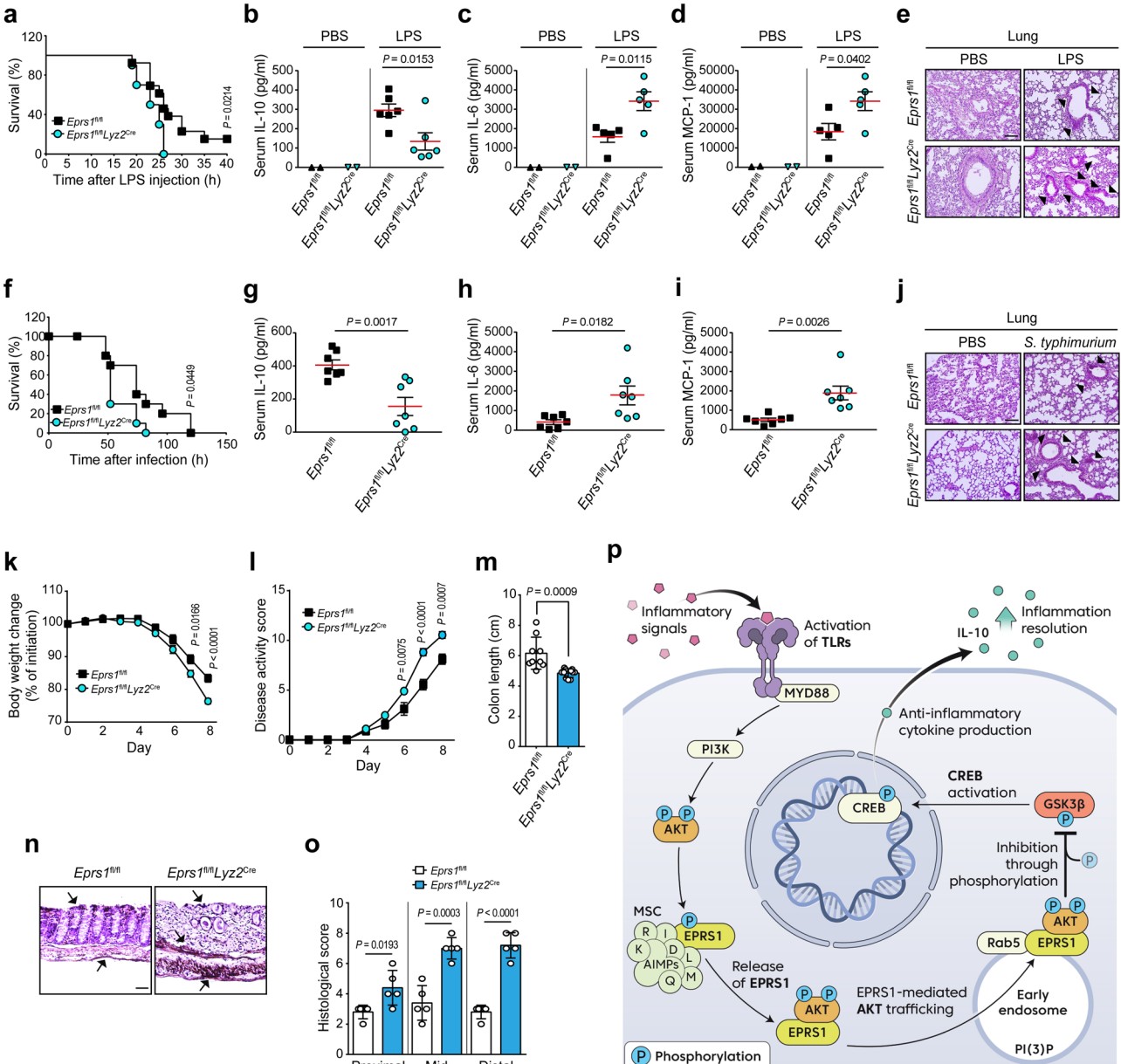

**Fig. 6 | EPRS1 protects mice from inflammatory insults. a** Survival rates of *Eprs1*^fl/fl^ (*n* = 13) and *Eprs1*^fl/fl^*Lyz2*^Cre^ (*n* = 10) mice were examined following intraperitoneal injection of LPS (30 mg/kg). **b** Serum concentrations of IL-10 in *Eprs1*^fl/fl^ (*n* = 6) and *Eprs1*^fl/fl^*Lyz2*^Cre^ (*n* = 6) mice measured 6 h after LPS injection. **c, d** Concentration of IL-6 (**c**) and MCP-1 (**d**) in *Eprs1*^fl/fl^ (*n* = 5) and *Eprs1*^fl/fl^*Lyz2*^Cre^ (*n* = 5) mice serum, measured 24 h after LPS injection. **e** Hematoxylin–eosin (H&E)-stained sections of lung tissues from *Eprs1*^fl/fl^ and *Eprs1*^fl/fl^*Lyz2*^Cre^ mice 24 h after LPS injection. Scale bar, 200 μm. Black arrows indicate infiltration by inflammatory cells and thickened alveolar walls. **f** Survival rates of *Eprs1*^fl/fl^ and *Eprs1*^fl/fl^*Lyz2*^Cre^ mice (*n* = 10 per group) were examined following intraperitoneal injection of *S. typhimurium* (5 × 10⁷ cfu). **g** Serum concentration of IL-10 in *Eprs1*^fl/fl^ (*n* = 7) and *Eprs1*^fl/fl^*Lyz2*^Cre^ (*n* = 7) mice, measured 6 h after *S. typhimurium* infection. **h, i** Concentration of IL-6 (**h**) and MCP-1 (**i**) in *Eprs1*^fl/fl^ (*n* = 7) and *Eprs1*^fl/fl^*Lyz2*^Cre^ (*n* = 7) mice serum, measured 24 h after *S. typhimurium* infection. **j** H&E-stained sections of lung tissues from *Eprs1*^fl/fl^ and

*Eprs1*^fl/fl^*Lyz2*^Cre^ mice 24 h after *S. typhimurium* infection. Scale bar, 200 μm. Arrows indicate infiltration by inflammatory cells and sites of damage. **k–m** Daily changes in body weight (**k**), disease activity score (**l**), and colon length (**m**) in *Eprs1*^fl/fl^ (*n* = 9) and *Eprs1*^fl/fl^*Lyz2*^Cre^ (*n* = 11) mice treated with 2% DSS. **n, o** Representative H&E staining (**n**) and histological scores (**o**) for colon dissected from *Eprs1*^fl/fl^ and *Eprs1*^fl/fl^*Lyz2*^Cre^ (*n* = 5 per group) mice on Day 8 post-treatment with DSS. Scale bar, 50 μm. Arrows indicate inflammatory cell infiltration and thickening of the colonic wall. **p** Proposed model of EPRS1-mediated early endosome-specific AKT anti-inflammatory signaling activation that helps to maintain physiological homeostasis during inflammation. See text for details. The data shown are representative of three independent experiments, each with similar results. All data are expressed as the mean ± SEM. *P*-values were calculated using the log-rank tests (**a**, **f**) and two-tailed unpaired *t*-tests (**b–d**, **g–i**, **k–m**, **o**). Source data are provided as a Source Data file.

bacterial infection, while IFNγ-mediated EPRS1 phosphorylation at Ser999 shapes translational fine tuning to protect cells from inflammation and injury in the presence of persistent inflammatory stimuli.

Because AKT acts as a central node in cell signaling downstream of growth factors, cytokines, and other cellular stimuli, AKT substrates that contribute to diverse cellular roles in promoting cell growth, proliferation, survival, metabolism, and motility

pathways must be regulated specifically to maintain homeostasis[16]. In this regard, endocytic platform-mediated signaling provides signaling specificity and insulation[72,73]. A representative example of this is APPL1/AKT signaling[22]: upon growth factor stimulation, the effector protein APPL1 partitions AKT into endosomal compartments to ensure selective regulation of its substrate GSK3β for specific cell survival signaling[22]. Although APPL1 is a multi-

functional endosomal signaling effector protein[74], we observed no association of APPL1 with EPRS1, at least in the context of anti-inflammatory endosomal AKT signaling (data not shown), suggesting that EPRS1 is a selective effector for anti-inflammatory endosomal AKT/GSK3β signaling.

PTEN is distributed mainly along microtubules, which are tethered to vesicles via PI(3)P throughout the cytoplasm[23]. The PTEN CBR3 loop contains a lipid-binding sequence specific for PI(3)P; the lipid interaction targets PTEN to endosomal membranes, where it terminates PI3K/AKT signaling. Termination of signaling is accomplished by the lipid phosphatase activity of PTEN to PIP3 on PI(3)P-positive endocytic vesicles that interact with plasma membrane signal activation-mediated internalized PIP3-containing membranes[23]. EPRS1 contributes to maintaining AKT phosphorylation by prolonging transient activation of PIP3-bound AKT in intracellular vesicles (Figs. 4g and 5a). It is assumed that inflammatory signal-induced activated, PIP3-associated AKT is internalized to phosphorylate MSC-associated EPRS1, which in turn interacts with PI(3)P-positive endosomal compartments[23]. The activity of EPRS1 is proportional to the level of sustainable inflammatory stimuli. This unique feature of EPRS1 might be critical for maintaining immunological homeostasis for extended periods of time, and also might provide clues regarding sustainable AKT activation in the cytosol. EPRS1 shares the PI(3)P lipid-binding sequence (Fig. 4b) found in the PTEN CBR3 loop. Future studies should explore whether EPRS1 competes with PTEN endosomes to dictate AKT signal strength under inflammatory conditions. Meanwhile, prolonged EPRS1-mediated AKT activation must be terminated in a timely manner. It would also be valuable to investigate whether PTEN controls EPRS1-mediated endosomal AKT signaling.

Most AKT substrates are phosphorylated and functionally regulated by all AKT isoforms: AKT1, AKT2, and AKT3. Nonetheless, many substrates are uniquely targeted by a specific AKT isoform[16]. AKT1 is ubiquitously expressed, whereas AKT2 is expressed in insulin-responsive tissues including adipocytes and skeletal muscles[39]. Expression of AKT3 is more restricted and is largely detected in the brain and testes[75]. AKT1 and AKT3 exhibit a preference for plasma membrane PIP3, whereas AKT2 specifically localizes to endomembranes for up to 30 min after growth factor stimulation[39]. Several lines of evidence also support the location of AKT1 in early endosomes upon transfection or in activated immune cells[76,77]. Moreover, AKT1 has been implicated in the regulation of TLR-induced macrophage activation and regulation of inflammation[8,78], whereas AKT2 preferentially regulates glucose and lipid metabolism[79,80]. For example, APPL1 directly interacts with AKT2, but insulin treatment dissociates the APPL1–AKT2 complex, which in turn promotes GLUT4 membrane trafficking in adipocytes under insulin signaling[81–83]. Collectively, although the data suggest that EPRS1 interacts with all AKT isoforms (Fig. 3e), we cannot rule out the possibility that isoform specificity may be regulated in a spatiotemporal and tissue-type-specific manner under certain inflammatory conditions.

Considering their broad regulatory activities, encompassing transcription, translation, and other cellular functions, ARSs may be mediators that maintain homeostasis in the face of diverse pathological challenges. Our bodies often confront infection and injury, and inflammation is a vital part of the immune system's response to those challenges; however, because it has potentially deleterious effects over the long term, inflammation must be kept under control. In that sense, the function of the ubiquitous housekeeping protein EPRS1 as an effector protein that coordinates specific AKT signaling to negatively regulate inflammation is consistent with the non-canonical roles of ARSs in maintaining homeostasis.

## Methods

All animal experiments were approved by the Institutional Animal Use and Care Committee of the Korea Research Institute of Bioscience and Biotechnology and were performed in accordance with the Guide for the Care and Use of Laboratory Animals (published by the US National Institutes of Health).

### Plasmid construction

Several EPRS1 constructs bearing FLAG, Strep, YFP, and GFP tags were generated. Briefly, a series of PCR-amplified truncated EPRS1 mutants harboring each domain were subcloned into GFP-, FLAG- or His-Sumo-tag–containing vectors. A Phosphomimetic mutant of EPRS1 (S886D/S999D) and a phosphorylation-resistant mutant (S886A/S999A) were generated. An endosome-association mutant of the EPRS1 (K974A/K977A/P978A/K980A) was also generated. Specific HA-, Strep-, GST-, mScarlet-, and FLAG-tagged full-length and truncated AKT mutants were generated. Rab5, Rab7, LAMP1, and LC3 were cloned into mCherry-tag vectors, and the active form of Rab5 (Q79L) was cloned into CFP-, mCherry-, and Strep-tag vectors. Primer sequences for site-directed mutagenesis are listed in Supplementary Table 2.

### Antibodies and reagents

The following primary antibodies were used for immunoblotting: anti-EPRS1 (1:2000, Abcam, ab31531; 1:1000, Thermo Fisher Scientific, A303-957A), anti-MARS1 (1:1000, Abcam, ab50793), anti-AIMP3 (1:1000, Neomics, NMS-01-0002), and anti-KARS1 (1:1000, Neomics, NMS-02-0005). The following antibodies were from Cell Signaling Technology: anti-FLAG (1:2000, #8146), anti-HA (1:2000, #2999), anti-Actin-HRP (1:5000, #12620), anti-AKT (1:2000, #4691), anti-AKT1 (1:1000, #2967), anti-AKT2 (1:1000, #5239), anti-p-AKT Thr308 (1:2000, #2965), anti-p-AKT Ser473 (1:2000 for immunoblotting; 1:200 for confocal analysis, #4060), anti-Rab5 (1:1000, #3547), anti-GSK3β (1:1000, #12456), anti-p-GSK3β Ser9 (1:1000, #5558), anti-CREB (1:1000, #9197), anti-p-CREB Ser133 (1:1000, #9198), anti-NF-kB p65 (1:1000, #8242), anti-p-NF-kB p65 Ser536 (1:1000, #3033), anti-mTOR (1:500, #2983), anti-p-mTOR S2448 (1:500, #5536), anti-S6K (1:1000, #2708), anti-p-S6K Thr389 (1:1000, #9234), anti-ERK1/2 (1:1000, #4695), anti-ERK1/2 T202/Y204 (1:1000, #9101), anti-JNK (1:1000, #9252), anti-p-JNK T183/Y185 (1:1000, #4668), anti-p38 (1:1000, #9212), anti-p-p38 T108/Y182 (1:1000, #4511), anti-RSK1 (1:1000, #9333), anti-PKC (1:1000, #59754), and anti-GST (1:2000, #2622). The following antibodies were from Santa Cruz Biotechnology: anti-STK26 (1:1000, sc-376649), anti-SYK (1:1000, sc-1240), anti-OSR1 (1:1000, sc-376545), anti-PKA (1:1000, sc-365615), anti-PKN (1:1000, sc-393344), anti-BTK (1:1000, sc-81735), and anti-GFP (1:1000, sc-9996). Other antibodies included anti-Strep (1:10,000, IBA, #2-1509-001, GmbH, Germany), Alexa Fluor 594-conjugated anti-rabbit IgG (1:200, Invitrogen, A11037), and anti-p-EPRS1 Ser990 (1:500; Abclon, Korea)[30]. Phospho-specific Ser886 (1:1000) and Ser999 (1:1000) antibodies[26] were provided by Prof. Paul L. Fox (Cleveland Clinic Lerner Research Institute). The Screen-Well kinase inhibitor library (Enzo Life Sciences, BML-2832-0100) was used for kinase profiling. Other reagents and materials included LPS (Sigma, L2630), Bacterial lipoprotein Pam3CSK4 (BLP, Invivogen, tlrl-pms), zymosan (Invivogen, tlrl-zyn), poly(I:C) (Invivogen, tlrl-pic), ODN2395 (CpG, Invivogen, tlrl-2395), IFNγ (KOMA, Korea, K0921033), a Proteome Profiler Mouse Cytokine Array Kit (R&D Systems, ARY006), protein A/G PLUS-agarose beads (Santa Cruz Biotechnology, sc-2003), GST resin (GE Healthcare, #17-0756-01), FLAG M2 affinity gel (Sigma, A2220), Strep resin (IBA, #2-1201-025), Ni-NTA agarose beads (Qiagen, #30230), GFP-Trap agarose (ChromoTek, gta-100), and a Glutathione–Sepharose 4B column (GE Healthcare, #17-0756-01). Transfection was performed using Lipofectamine 2000 (Thermo Fisher Scientific, #11668030) and the TurboFect (Thermo Fisher Scientific, R0531) reagents.

## Cell culture

293T (ATCC), HeLa (ATCC), and Raw 264.7 (KCLB) cells were cultured in DMEM medium (Gibco) supplemented with 10% fetal bovine serum (FBS; Gibco), and 1% antibiotic–antimycotic (Gibco). U937 (ATCC) cells were cultured in RPMI 1640 medium (Gibco) supplemented with 10% FBS and 1% antibiotic–antimycotic. 293 T/TLR2 (Invivogen) and 293T/TLR4 (Invivogen) were grown in DMEM containing 10% FBS, 1% antibiotic–antimycotic, 100 µg/ml Normocin (Invivogen), and 1X HEK Blue™ selection (Invivogen). All cells were incubated at 37 °C in a humidified 5% CO₂ environment and confirmed to be free of mycoplasma. For inflammatory stimuli, cells were pre-incubated in a serum-reduced medium (2% FBS) for 4 h and treated with TLR ligands for indicated times without changing the serum level to exclude any extra effects derived from growth factors and nutrients. Cells were infected with vesicular stomatitis virus (VSV-GFP, MOI = 1) for 2 h in reduced serum (1% FBS)-containing medium.

## Real-time RT-PCR

Total RNA was extracted from cells using the RNeasy RNA extraction Mini-Kit (QIAGEN). Purified RNA was treated with RNase-free DNase at 37 °C for 30 min. Quantitative PCR was performed using gene-specific primer sets and GoTaq qPCR Master Mix (Promega, A6002). Real-time PCR was carried out using a LightCycler 96 (Roche Diagnostics) according to the manufacturer's instructions. Data were normalized against glyceraldehyde-3-phosphate dehydrogenase expression, and relative expression was calculated using the ΔΔCT method. Primer sequences are listed in Supplementary Table 2.

## Enzyme-linked immunosorbent assay

ELISA was performed to detect cytokines and chemokines in mouse sera or cell culture supernatants. Mouse IL-6, TNF-α, MCP-1, IL-10 (BD Biosciences), and IFNγ (KOMA, Korea) were used for analysis.

## Dual-luciferase reporter assay

293T, 293/hTLR2, or 293/hTLR4 cells were transfected with a mixture containing a luciferase reporter plasmid, a *Renilla* luciferase internal control vector (phRL-TK; Promega, E2810), and each of the indicated plasmids. The reporter gene assay was performed 24 h post-transfection using a luminometer (Promega) and the dual-luciferase reporter assay system (Promega, E1500). Data are expressed as relative firefly luciferase activity normalized against *Renilla* luciferase activity.

## Immunoblotting and immunoprecipitation analysis

For immunoblot analysis, cells were lysed with RIPA lysis buffer (20 mM Tris–HCl pH 7.4, 150 mM NaCl, 1% NP-40, and 1 mM EDTA) containing protease inhibitor cocktail (GenDepot, P3100-001) and phosphatase inhibitor cocktail (GenDepot, P3200-001). Whole-cell lysates were subjected to SDS-PAGE followed by immunoblotting with the indicated antibodies. To detect phosphorylated EPRS1 proteins, cells were lysed with Phosphosafe extraction buffer (Millipore, #71296) containing protease inhibitor at 4 °C. Immunoprecipitation was performed with cell lysates that had been pre-cleared with protein A/G beads for 1 h at 4 °C. The pre-cleared cell lysates were incubated overnight with the indicated antibodies at 4 °C, followed by incubation with 30 µl protein A/G PLUS-agarose beads for 3 h at 4 °C. The immunoprecipitates were collected and washed extensively with lysis buffer before immunoblot analysis.

## Expression and purification of recombinant proteins

Plasmids expressing Strep-tagged EPRS1 were transfected into 293T cells. Cells were lysed with lysis buffer (20 mM Tris–HCl pH 8.0, 1% NP-40, 150 mM NaCl, 1 mM EDTA, and protease inhibitor cocktail) and lysates were centrifuged at 16,000×g for 20 min at 4 °C. Cell lysates containing Strep-tagged EPRS1 were applied to a Strep-Tactin Superflow column (IBA) and washed with Strep wash buffer (100 mM HEPES pH 8.0, 150 mM NaCl, and 1 mM EDTA). The proteins were eluted with Strep elution buffer (Strep wash buffer containing 2.5 mM desthiobiotin). A plasmid expressing His-Sumo-tagged Linker (aa 683–1023) was transformed into *Escherichia coli* BL21-CodonPlus (DE3)-RIPL cells and expression was induced by treatment with 0.5 mM IPTG at 18 °C for 18 h. The harvested cells were suspended in buffer A (50 mM Tris–HCl pH 7.5, 150 mM NaCl) and lysed by sonication on ice. Cell lysates were centrifuged at 25,000×g for 1 h at 4 °C. Supernatants containing His-Sumo-tagged EPRS1 Linker were loaded onto a Ni-NTA agarose column, washed extensively with buffer A, and eluted with 250 mM imidazole. All eluted proteins were separated by SDS–PAGE, followed by staining with Coomassie Brilliant Blue.

## Mass spectrometry

293T cells were transfected with pEXPR-IBA105 (empty vector, vehicle) or Strep-EPRS1 plasmids under unstimulated (−LPS) or stimulated (+LPS) condition and lysed in lysis buffer containing 1% NP-40, 20 mM Tris–HCl pH 8.0, 150 mM NaCl, 2 mM EDTA, 10% glycerol, and protease inhibitor cocktail. Lysates were centrifuged at 16,000×g for 20 min at 4 °C. Cell lysates containing Strep-tagged EPRS1 were applied to a Strep-Tactin Superflow column (IBA) for affinity purification and washed with Strep wash buffer (100 mM HEPES pH 8.0, 150 mM NaCl, and 1 mM EDTA), and proteins were eluted with Strep elution buffer containing 2.5 mM desthiobiotin. Eluted proteins were separated by SDS–PAGE and excised into five gel pieces. Individual gel pieces were destained and subjected to in-gel digestion using trypsin, and digested peptides were analyzed using an Orbitrap Exploris 240 (Thermo Fisher Scientific) connected to an RSLCnano u3000 system (Thermo Fisher Scientific) equipped with an autosampler. Dried peptide samples were resuspended in 12 µl of 0.1% formic acid/2% acetonitrile, and an aliquot was injected onto a reversed-phase peptide trap EASY-Column (length = 2 cm, internal diameter = 75 µm, 3 µm, acclaim pepmap 100, Thermo Fisher Scientific) and a reversed-phase analytical Column (length = 25 cm, internal diameter = 75 µm, 3 µm, BEH300, Waters). The duration of the LC gradient was 120 min. Peptides were eluted using a linear gradient of 2–20% buffer B over 100 min, 20–32% buffer B over 20 min (buffer A = 0.1% formic acid in H₂O, buffer B = 0.1% formic acid in acetonitrile) at a flow rate of 300 nl/min. The Orbitrap Exploris 240 mass analyzer was operated in positive ESI mode using HCD to fragment peptides following separation by HPLC. The temperature and voltage applied to the capillary were 275 °C and 1.9 kV, respectively. All data were acquired with the mass spectrometer operating in automatic data-dependent switching mode. MS spectra were scanned from 350 to 1200 *m/z* with a resolution of 120,000. The automatic gain control (AGC) target was set at 3,000,000 ions with a maximum fill time of 50 ms. A total of 15 data-dependent MS/MS scans were selected using an isolation window of 2.0 *m/z*, a standard AGC target, a maximum fill time of 200 ms, and normalized collision energy of 30. Dynamic exclusion was performed with a repeat count of 1, exclusion duration of 20 s. The minimum MS ion count for triggering MS/MS was set at 100,000 counts.

## AKT kinase assay

Total cell lysates were pre-cleared with blank beads and then incubated overnight with an anti-AKT antibody, followed by incubation with a 50% protein A/G bead slurry for 3 h. After five washes (the last wash was performed using kinase buffer), the immune complexes were incubated with full-length EPRS1or the Linker, along with the kinase reaction mix from the AKT Activity Assay kit (Abcam, ab65786). Phosphorylation of EPRS1 was analyzed by western blotting with an antibody against phospho-Ser999 EPRS1.

## RNAi

Cells were transfected with duplex siRNA using Lipofectamine 2000 (Thermo Fisher Scientific). A non-targeting siRNA was used as a control. Cells were incubated with siRNA or control for 24–36 h before exposure to LPS. The siRNA sequences are listed in Supplementary Table 2.

## Confocal microscopy

Cells were seeded onto eight-chamber slides, followed by the indicated treatment or transfection. The cells were fixed for 10 min in 4% paraformaldehyde at room temperature and then permeabilized by incubation (3 × 5 min) in HEPES buffer (100 mM HEPES pH 8.0, 150 mM NaCl) containing 0.1% Triton X-100[84]. The fixed cells were blocked for 1 h with HEPES buffer containing 10% FBS and then incubated overnight at 4 °C with appropriate primary antibodies. After five washes with HEPES buffer, the cells were incubated for 1 h at room temperature with appropriate secondary antibodies. After washing five times with HEPES buffer, the cells were stained with Hoechst (Invitrogen, H3570) for 10 min. Live cells were imaged at 37 °C/5% CO$_2$ (Live Cell Instruments) using a Chamlide TC system placed on a microscope stage. Images were acquired under a Nikon laser scanning confocal microscope (C2plus, A1R) and analyzed using the NIS-Elements software.

## Membrane lipid overlay assay

Echelon Biosciences PIP Strips (product number: P-6001) were incubated overnight at 4 °C in blocking buffer (20 mM Tris–HCl pH 8.0, 150 mM NaCl, 0.1% Tween-20, and 3% BSA). After the blocking buffer was discarded, the membranes were incubated with purified proteins (50 µg/ml) for 1 h at room temperature with gentle agitation. To detect the purified Strep-EPRS1, the membranes were incubated with an HRP-conjugated anti-Strep antibody. The blots were developed using a chemiluminescence substrate solution (Amersham ECL Prime, GE Healthcare, RPN2232).

## Early endosome isolation

Subcellular fractionation was performed as previously described[85]. Specifically, U937 cells were stimulated (or not) with LPS (100 ng/ml) for 30 min at 37 °C. Then, the cells were harvested and incubated with homogenization buffer (250 mM sucrose, 1 mM EDTA, protease inhibitor, and phosphatase inhibitor), in which cells were gently lysed by passing through a syringe needle (50 times on ice). After serial centrifugation (1000×$g$ followed by 10,000×$g$), the post-nuclear supernatant was collected and adjusted to 1 ml total volume. Next, 2.4 ml of 45% sucrose was transferred to the bottom of a swing bucket tube and overlaid with 4.8 ml of 35% sucrose, 3 ml of 25% sucrose, and 1 ml of sample. Following ultracentrifugation, for 100,000×$g$ for 1 h (Hitachi Himac P40ST swing rotor), 1.5 ml fractions were collected from top to bottom to detect endosomal markers.

## Isolation of macrophages from mice

Mice were injected intraperitoneally with 3 ml of 3% thioglycollate (Sigma, B2551) in sterile PBS once daily for 3 days, and sacrificed by cervical dissociation. The cells were harvested by washing the peritoneal cavities with 3 ml sterile cold PBS and centrifugation of their peritoneal lavage fluid at 800×$g$ at 4 °C for 5 min. The pelleted peritoneal macrophages were resuspended and used for subsequent studies after suitable dilution. BMDMs were isolated from 6- to 8-week-old mice, and red blood cells were lysed with ammonium–chloride–potassium (ACK) lysing buffer (Gibco/Thermo Fisher Scientific, A1049201). BMDMs were cultured in DMEM containing 10% FBS and GM-CSF (R&D Systems, 415-ML-010) for 6–7 days.

## Mouse experiments

Mice with conditional deletion of C57BL/6N EPRS1 in macrophages were generated using LysM promoter-driven, Cre recombinase-mediated excision of exons of the EPRS1 gene. Sex-matched LysMCre−negative homozygous floxed EPRS1 littermates were used as controls. Mice were intraperitoneally treated with LPS (30 mg/kg) or infected with *S. typhimurium* (5 × 10$^7$ colony-forming units) to monitor survival rates. To investigate the serum cytokine levels, the sample was harvested at 6 or 24 h after LPS treatment or *S. typhimurium* infection. Mouse lung, liver, and kidney tissues after LPS treatment or mouse lung tissues after *S. typhimurium* infection were harvested for histological analysis at 24 h. Acute colitis was induced by the administration of 2–5% (w/v) DSS (MP Biomedicals, #0216011080) in drinking water. Weight changes were calculated as the percent change in weight relative to the baseline weight. Disease activity score (range: 0–12) was calculated by summing up the scores[86,87] of body weight loss (0, none; 1, 1–5%; 2, 6–10%; 3, 11–15%; 4, over 15%), stool consistency (0, well-formed pellets; 2, loose stools; 4, diarrhea), and fecal blood (0, negative hemoccult test; 1, positive hemoccult test; 2, blood visibly present in the stool; 3, blood visibly present in the stool and blood clotting on the anus; 4, gross bleeding). Colons were dissected and washed with PBS. Mice were maintained in a specific pathogen-free facility on a 12 h light/dark cycle at 22 ± 2 °C with free access to food and water. All mice were euthanized by CO$_2$ asphyxiation.

## Histology

The lung, kidney, liver, and colon tissue samples were harvested and fixed in 10% neutral buffered formalin overnight at room temperature. Fixed tissues were washed for over 8 h with tap water and embedded in paraffin blocks on a Tissue Embedding Center module (Tissue-Tek, Sakura Finetek Co., Japan). Tissue slides were prepared at a thickness of 7 µm using a microtome (Accu-Cut, Sakura Fintek Co.). Tissue sections were deparaffinized in xylene, rehydrated in ethanol, and stained with hematoxylin and eosin (H&E) according to the standard protocol. Stained sections were imaged under an optical microscope (BX51, Olympus Corp, Tokyo, Japan). The histological score was determined as described previously[88] by summing up the scores (range: 0–8) of epithelial damages (0, normal morphology; 1, loss of goblet cells; 2, loss of goblet cells in large areas; 3, loss of crypts; 4, loss of crypts in a large area) and immune cell infiltrations (0, no infiltration; 1, infiltrate around crypt basis; 2, infiltrate reaching to muscularis mucosae; 3, extensive infiltration; 4, infiltration of the submucosa).

## Statistical analysis

Statistical analyses were performed using Prism 6 (GraphPad). Data were analyzed using Student's unpaired *t*-test, the log-rank test, or the non-parametric Mann–Whitney test as appropriate. Data are expressed as means ± SEM unless stated otherwise, and all experiments were repeated at least three times. Gene ontology terms were largely classified into the overarching protein analysis through evolutionary relationships (PANTHER) GOSlim terms[89].

## Reporting summary

Further information on research design is available in the Nature Research Reporting Summary linked to this article.

# Data availability

All data are available within this article and the Supplementary Information files. The mass spectrometry proteomics data generated in this study have been deposited in the ProteomeXchange Consortium via the PRIDE[90] partner repository under accession code PXD036072. Source data are provided with this paper.

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

## Acknowledgements

We thank all members of the Infection and Immunity Research Laboratory at the Korea Research Institute of Bioscience and Biotechnology (KRIBB) for technical assistance and helpful discussions. We would like to thank Dr. Won Do Heo at KAIST for his helpful advice on confocal microscopy experiments. This work was supported by a Samsung Science and Technology Foundation grant (SSTF-BA1902-04), the National Research Council of Science & Technology (No. CAP20013-000), and the National Research Foundation of Korea (NRF-2021M3A9I4022934) grants funded by MSIT, and the KRIBB Initiative Program to M.H.K.

## Author contributions

Conceptualization, E.-Y.L. and M.H.K.; formal analysis, E.-Y.L., S.-M.K., J.H.H., S.Y.J., S.P., S.C., G.S.L., J.H., J.H.M., P.L.F., S.K., C.-H.L., and M.H.K.; funding acquisition, M.H.K.; investigation, E.-Y.L., S.-M.K., J.H.H., S.Y.J., S.P., S.C., and G.S.L.; supervision, M.H.K.; validation, E.-Y.L., S.-M.K., C.-H.L., and M.H.K.; writing, E.-Y.L. and M.H.K.

## Competing interests

The authors declare no competing interests.
