## [Peer Review File · Nature Communications]

Glutamyl-prolyl-tRNA synthetase 1 coordinates early endosomal anti-inflammatory AKT signalingReviewer #1 (Remarks to the Author):

In the present report xx et al demonstrated the role of EPRS1 in regulating anti-inflammatory responses in macrophages. They showed that upon TLR triggering EPRS was phosphorylated by AKT, in an AKT isoform-independent manner. They also showed that the Akt localized in early endosomes in complex with EPRS1 and Rab5. Moreover, they showed that association of EPRS with Akt is essential to mediate the anti-inflammatory actions of Akt. In vivo, lack of EPRS1 resulted in exacerbated inflammatory response to LPS, salmonella infection or inflammatory bowel disease. The results are novel and important in the field and are in line with data available in the literature and unpublished information that we have in our lab. All experiments are appropriately controlled and of very good quality.

Comments:

1. Expression levels of EPRS1 in EPRS1 +/- BMDMs should be presented, in view of the data in figure 1.
2. In Supplemental figure 1 the authors should make clear that the concentration noted is for EPRS-expressing plasmid and not protein. This should be noted on the graphs.
3. In Figure 3 the authors should show on the figure which Akt isoform was expressed in each case in the tagged constructs.
4. Data shown in figure 6p and supp 8f and 8g that demonstrate reduced Akt phosphorylation is not clear if this is due to reduced Akt activation and not to different cell composition in the tissue. Contribution of EPRS1 in Akt activation was shown in the cell culture models and these figures may be misleading if Akt phosphorylation is not shown in a cell-specific setting. Therefore these figures could be removed.
5. A figure summarizing the role of EPRS1 and Akt in the context of inflammatory signaling as part of the discussion may be useful to the reader.
6. Is it possible that translocation of EPRS1 to endosomes and dissociation from the ARS complex may affect protein synthesis and therefore reduce translation of pro-inflammatory mediators? The potential effect on protein translation should be discussed.

Reviewer #2 (Remarks to the Author):

Overall, this is a potentially interesting study. My comments are only central to the inflammation components of the manuscript - Figure 6 and Supplementary Figs. 7 and 8.

1. The rationale/justification for using the LPS-induced model of inflammation, the Salmonella colitis model, and the DSS colitis model needs to be better articulated and contextualized.
2. Figure 6. The Western blots should be quantitated via densitometry with statistics demonstrated.
3. The disease severity score and rubric needs to be described in detail.
4. These comments are focused comments that are relevant to the data presented in Figures 7 and 8 - and need to be addressed, as well.

Description of changes to the manuscript:

1. Reviewer #1:

(Remarks to the Author)

In the present report xx et al demonstrated the role of EPRS1 in regulating anti-inflammatory responses in macrophages. They showed that upon TLR triggering EPRS was phosphorylated by AKT, in an AKT isoform-independent manner. They also showed that the Akt localized in early endosomes in complex with EPRS1 and Rab5. Moreover, they showed that association of EPRS with Akt is essential to mediate the anti-inflammatory actions of Akt. In vivo, lack of EPRS1 resulted in exacerbated inflammatory response to LPS, salmonella infection or inflammatory bowel disease. The results are novel and important in the field and are in line with data available in the literature and unpublished information that we have in our lab. All experiments are appropriately controlled and of very good quality.

Response: We sincerely thank the Reviewer for taking the time to review our manuscript and for providing positive and constructive feedback.

Comments:

Q1. Expression levels of EPRS1 in EPRS1 +/- BMDMs should be presented, in view of the data in figure 1.

Response: As suggested, we now show the mRNA and protein expression levels of EPRS1 in BMDMs isolated from *Eprs1^{+/+}* and *Eprs1^{+/-}* mice in the revised manuscript (Figure 1a, b).

Q2. In Supplemental figure 1 the authors should make clear that the concentration noted is for EPRS-expressing plasmid and not protein. This should be noted on the graphs.

Response: We thank the Reviewer for this thoughtful comment. We have clarified this issue by changing 'EPRS1' to 'EPRS1-FLAG' in the graphs of Supplementary Figure 1a-c. We have also specified the concentrations of EPRS1-expressing plasmid in the legend of Supplementary Figure 1.

Q3. In Figure 3 the authors should show on the figure which Akt isoform was expressed in

each case in the tagged constructs.

Response: We appreciate this pertinent comment. We have corrected 'AKT' to 'AKT1' isoform in Figure 3a-c in the revised manuscript. Since we used AKT (pan) antibody (Cell Signaling Technology, Cat#4691) for immunoprecipitation, the 'AKT' label in Figure 3f has not been changed in the revised manuscript.

Q4. Data shown in figure 6p and supp 8f and 8g that demonstrate reduced Akt phosphorylation is not clear if this is due to reduced Akt activation and not to different cell composition in the tissue. Contribution of EPRS1 in Akt activation was shown in the cell culture models and these figures may be misleading if Akt phosphorylation is not shown in a cell-specific setting. Therefore these figures could be removed.

Response: We thank the Reviewer for providing this insightful comment. Although tissue-resident local or recruited macrophages exist in mouse lung and liver, tissues are definitely composed of multiple and heterogeneous cell compositions. We tried to evaluate the systemic effects of EPRS1-mediated AKT regulation *in vivo*. However, we cannot rule out the possibility raised by the Reviewer. Thus, as suggested, we have removed the data (Fig. 6p and Supplementary Fig. 8f, g) from the revised manuscript.

Q5. A figure summarizing the role of EPRS1 and Akt in the context of inflammatory signaling as part of the discussion may be useful to the reader.

Response: As suggested, we have added a figure summarizing the role of EPRS1 and AKT in the context of inflammatory signaling (Fig. 6p) together with a description at the end of Results section in the revised manuscript.

Q6. Is it possible that translocation of EPRS1 to endosomes and dissociation from the ARS complex may affect protein synthesis and therefore reduce translation of pro-inflammatory mediators? The potential effect on protein translation should be discussed.

Response: Response: We thank the Reviewer for this thoughtful comment. It is well-known that the cytoplasmic MSC serves as a reservoir for ARSs, which can respond rapidly to cellular stresses without the need for *de novo* transcription and translation^{1, 2}. Although the fundamental function of the MSC is not well understood, it is known that *in vivo* depletion of the AIMP2 (previously named p38), the known scaffold protein required for the assembly of

the MSC, does not affect global protein synthesis and cell growth³, and only a small portion of the components is released from the MSC to perform non-catalytic regulatory activities in a stimulus-dependent manner while not reducing overall protein synthesis⁴. In this regard, a recent report demonstrated that there is no change in global protein synthesis and cell growth even in the absence of arginyl-tRNA synthetase (RARS1) and glutaminyl-tRNA synthetase (QARS1) in the MSC⁵. This study further investigated whether exclusion of RARS1 and QARS1 from the MSC hinders mRNA translation, and showed that the levels of newly synthesized proteins are well maintained in cells lacking MSC-localized RARS1 and QARS1⁵. EPRS1 released from the MSC preserves its translational role and does not disrupt total protein synthesis^{2, 6, 7}. Based on these previous results, we believe that the release of inflammatory signal-specific EPRS1 from the MSC would not reduce mRNA translation. Thus, it is unlikely that EPRS1-mediated anti-inflammatory immune activation is linked to a decrease in the translation of pro-inflammatory mediator genes. As suggested by the Reviewer, we have discussed these potential points on protein translation in the Discussion section of the revised manuscript. In fact, EPRS1 is specifically involved in pro-inflammatory transcript-selective translational control. Upon interferon- γ (IFN γ) activation, EPRS1 is released from the MSC and acts as a component of the IFN γ -activated inhibitor of translation (GAIT) complex that binds to the 3' UTR GAIT element in multiple pro-inflammatory transcripts and represses their translation in human myeloid cells. This was described in the Discussion section of the original version of the manuscript.

Reviewer #2:

(Remarks to the Author)

Overall, this is a potentially interesting study. My comments are only central to the inflammation components of the manuscript - Figure 6 and Supplementary Figs. 7 and 8.

Response: We sincerely thank the Reviewer for taking the time to review our manuscript and for providing constructive feedback.

Q1. The rationale/justification for using the LPS-induced model of inflammation, the Salmonella colitis model, and the DSS colitis model needs to be better articulated and contextualized.

Response: We greatly appreciate the Reviewer's insightful comments. In this study, we explored the role of EPRS1 during inflammatory conditions. Therefore, we used representative

animal models under inflammatory conditions to evaluate the anti-inflammatory function of EPRS1. LPS is one of the major bacterial products and pathogenic mediators leading to inflammation. LPS-mediated endotoxic shock in mice is characterized by severe systemic inflammation, in which macrophages play an important role in LPS responses⁸. Infection with pathogenic bacteria including *Salmonella* is a key factor that can influence the development of inflammatory disease^{9, 10}. Lastly, it is well known that bacteria are key factors causing intestinal inflammation in patients with IBD¹¹. Each model possesses unique features and has an array of advantages and disadvantages for the study of inflammatory diseases¹². Therefore, we used three mouse models together to explore the roles of EPRS1 in a context of a complex immune response to irritants, bacteria-derived products, a pathogen, and tissue damage. We have also referred to several studies that used similar inflammatory models^{13, 14, 15}. Especially, a recent study used the same mouse model set to investigate the regulatory functions of ASB1 protein in inflammation¹⁶. As suggested, we have revised our manuscript by including a rationalized description of the mouse models used and related references.

Q2. Figure 6. The Western blots should be quantitated via densitometry with statistics demonstrated.

Response: We thank the Reviewer for this comment. The other Reviewer also commented on this data along with Supplementary Fig. 8f and 8g. He/she suggested that we delete the *in vivo* AKT phosphorylation data because it is not clear whether the reduced AKT phosphorylation in EPRS1-deficient BMDMs from *Eprs1^{fl/fl}Lyz2^{Cre}* or *Eprs1^{+/-}* mice is caused by reduced AKT activation or by differences in the tissue cell composition. While tissue-resident local or recruited macrophages exist in mouse lung and liver, tissues also have multiple and heterogeneous cell compositions. We tried to evaluate the systemic effects of EPRS1-mediated AKT regulation *in vivo*. But, we could not rule out the possibilities that the other Reviewer raised and thus decided to delete those data (Fig. 6p, Supplementary Fig. 8f, g), as the other Reviewer suggested, from the revised manuscript. We sincerely hope that the Reviewer accepts these corrections.

Q3. The disease severity score and rubric needs to be described in detail.

Response: We deeply appreciate the Reviewer for providing us with the opportunity to correct and improve our manuscript. As requested, we now describe in detail the methods used to calculate the severity score (disease activity score) and histological score, together with

corresponding references, in the Methods section of the revised manuscript.

Q4. These comments are focused comments that are relevant to the data presented in Figures 7 and 8 - and need to be addressed, as well.

Response: We have corrected the indicated Figures in the revised manuscript. As described in Response to Q2, we have removed Fig. 6p and Supplementary Fig. 8f and 8g from the revised manuscript. Because IBD is a complicated inflammatory disease in which both adaptive and innate immune cells are involved¹⁷, we first monitored the anti-inflammatory effects of EPRS1 in heterozygous *Eprs1*^{+/-} mice, which revealed a significant difference in survival between heterozygous and wild-type mice (Supplementary Fig. 8a). Because we focused on the role EPRS1 in the context of macrophages, which also play a pivotal role in the pathogenesis of IBD¹⁸, we subsequently examined the anti-inflammatory effect of EPRS1 in myeloid-specific knockout mice, *Eprs1*^{fl/fl}*Lyz2*^{Cre}. Based on the rationale, we have rearranged these results in the revised manuscript: namely, Supplementary Fig. 7d (*Eprs1*^{fl/fl}*Lyz2*^{Cre}) has been changed to 'Supplementary Fig. 8f' following the results of heterozygous *Eprs1*^{+/-} mice (Supplementary Fig. 8a-e). Combined together, we think that these rearrangements have resulted in a clearer organization of the data of the inflammatory mouse models in the revised manuscript. We are deeply grateful to the Reviewer for his/her constructive suggestions on how to improve our manuscript.

References

1. Ray, P.S., Arif, A. & Fox, P.L. Macromolecular complexes as depots for releasable regulatory proteins. *Trends Biochem Sci* **32**, 158-164 (2007).
2. Lee, E.Y. *et al.* Infection-specific phosphorylation of glutamyl-prolyl tRNA synthetase induces antiviral immunity. *Nat Immunol* **17**, 1252-1262 (2016).
3. Kim, M.J. *et al.* Downregulation of FUSE-binding protein and c-myc by tRNA synthetase cofactor p38 is required for lung cell differentiation. *Nat Genet* **34**, 330-336 (2003).
4. Khan, K., Gogonea, V. & Fox, P.L. Aminoacyl-tRNA synthetases of the multi-tRNA synthetase complex and their role in tumorigenesis. *Transl Oncol* **19**, 101392 (2022).

5. Cui, H. *et al.* Regulation of ex-translational activities is the primary function of the multi-tRNA synthetase complex. *Nucleic Acids Res* **49**, 3603-3616 (2021).
6. Arif, A. *et al.* EPRS is a critical mTORC1-S6K1 effector that influences adiposity in mice. *Nature* **542**, 357-361 (2017).
7. Sampath, P. *et al.* Noncanonical function of glutamyl-prolyl-tRNA synthetase: gene-specific silencing of translation. *Cell* **119**, 195-208 (2004).
8. Fujihara, M. *et al.* Molecular mechanisms of macrophage activation and deactivation by lipopolysaccharide: roles of the receptor complex. *Pharmacol Ther* **100**, 171-194 (2003).
9. Schultz, B.M. *et al.* A Potential Role of Salmonella Infection in the Onset of Inflammatory Bowel Diseases. *Front Immunol* **8**, 191 (2017).
10. Mathur, R. *et al.* A mouse model of Salmonella typhi infection. *Cell* **151**, 590-602 (2012).
11. Zhang, Y.Z. & Li, Y.Y. Inflammatory bowel disease: pathogenesis. *World J Gastroenterol* **20**, 91-99 (2014).
12. Jiminez, J.A., Uwiera, T.C., Douglas Inglis, G. & Uwiera, R.R. Animal models to study acute and chronic intestinal inflammation in mammals. *Gut Pathog* **7**, 29 (2015).
13. Ahn, Y.H. *et al.* Secreted tryptophanyl-tRNA synthetase as a primary defence system against infection. *Nat Microbiol* **2**, 16191 (2016).
14. Xia, Y. *et al.* The macrophage-specific V-ATPase subunit ATP6V0D2 restricts inflammasome activation and bacterial infection by facilitating autophagosome-lysosome fusion. *Autophagy* **15**, 960-975 (2019).
15. Schultz, B.M. *et al.* Persistent Salmonella enterica serovar Typhimurium Infection Increases the Susceptibility of Mice to Develop Intestinal Inflammation. *Front Immunol* **9**, 1166 (2018).
16. Hou, P. *et al.* An unconventional role of an ASB family protein in NF-kappaB activation and

inflammatory response during microbial infection and colitis. *Proc Natl Acad Sci U S A* **118** (2021).

17. Danese, S. & Fiocchi, C. Ulcerative colitis. *N Engl J Med* **365**, 1713-1725 (2011).
18. Heinsbroek, S.E. & Gordon, S. The role of macrophages in inflammatory bowel diseases. *Expert Rev Mol Med* **11**, e14 (2009).

Reviewer #1 (Remarks to the Author):

The authors have addressed the comments raised by the reviewers.

Reviewer #2 (Remarks to the Author):

The authors have adequately addressed the concerns raised by this reviewer.

Reviewer #3 (Remarks to the Author):

I was asked to review the proteomics section of this paper, since others have already done an outstanding job of reviewing the scientific aspect of this paper. Likewise, it appears that the authors have done a good job of addressing these concerns. Thus, my comments will address the proteomics experiments.

First, the ProteomeXchange data should be made publicly available and the information published in the data availability section of the paper. Requiring a request to the author does not meet the current standards of publication of proteomics data.

In line 510, I believe there is a typo. Should it read "ratio of EPRS1 to actin", rather than ratio to action?

I believe that Figure 1b would be better as a bar chart rather than as a table. From the data in ProteomeXchange, it appears that replicates of each sample were run. Thus, these data can be presented as the RNAseq data were presented.

Description of changes to the manuscript:

1. Reviewer #1 (Remarks to the Author):

The authors have addressed the comments raised by the reviewers.

2. Reviewer #2 (Remarks to the Author):

The authors have adequately addressed the concerns raised by this reviewer.

3. Reviewer #3 (Remarks to the Author):

I was asked to review the proteomics section of this paper, since others have already done an outstanding job of reviewing the scientific aspect of this paper. Likewise, it appears that the authors have done a good job of addressing these concerns. Thus, my comments will address the proteomics experiments.

First, the ProteomeXchange data should be made publicly available and the information published in the data availability section of the paper. Requiring a request to the author does not meet the current standards of publication of proteomics data.

Response: We have provided the information of the proteomics data deposited in PRIDE with accession code PXD036072 in the Data availability section in the revised manuscript.

In line 510, I believe there is a typo. Should it read "ratio of EPRS1 to actin", rather than ratio to action?

Response: We thank the Reviewer for pointing out this typo. We have corrected the typo "action" to "actin" in the revised manuscript.

I believe that Figure 1b would be better as a bar chart rather than as a table. From the data in ProteomeXchange, it appears that replicates of each sample were run. Thus, these data can be presented as the RNAseq data were presented.

Response: We think that “Figure 1b” indicated by the Reviewer refers to “Figure 2b” in the manuscript. We apologize for causing confusion with the deposited proteomic data in ProteomeXchange. We generated the proteomic data with the samples prepared from 293T cells transfected with pEXPR-IBA105 (empty vector, vehicle) or Strep-EPRS1 plasmid under unstimulated (-LPS) or stimulated (+LPS) condition. Thus, the dataset contains “unstimulated vehicle (-LPS, Veh)” and “stimulated vehicle (+LPS, Veh)” as controls, and “unstimulated EPRS1 (-LPS, EPRS1)” and “stimulated EPRS1 (+LPS, EPRS1)”. We did not generate the data with replicates of each sample. We think that we made some confusion with sample labeling. Therefore, we have clarified labeling of those samples (Supplementary Table 1) and redeposited the dataset to ProteomeXchange via the PRIDE database with identifier PXD036072. The data can be accessed with reviewer account (Username: reviewer_pxd036072@ebi.ac.uk; Password: Please check the initial "Submission Complete" email for password). We have provided the accession code in the Data availability section in the revised manuscript. Accordingly, we think that we should leave “Figure 2b” as it is in the revised manuscript. We sincerely hope that the Reviewer accepts this notion.